# Adaptive and Optimal Second-order Optimistic Methods for Minimax Optimization

**Ruichen Jiang**
ECE department, UT Austin
rjiang@utexas.edu

**Ali Kavis**
ECE department, UT Austin
kavis@austin.utexas.edu

**Qiujiang Jin**
ECE department, UT Austin
qiujiangjin0@gmail.com

**Sujay Sanghavi**
ECE department, UT Austin
sanghavi@mail.utexas.edu

**Aryan Mokhtari**
ECE department, UT Austin
mokhtari@austin.utexas.edu

## Abstract

We propose adaptive, line search-free second-order methods with optimal rate of convergence for solving convex-concave min-max problems. By means of an adaptive step size, our algorithms feature a simple update rule that requires solving only one linear system per iteration, eliminating the need for line search or backtracking mechanisms. Specifically, we base our algorithms on the optimistic method and appropriately combine it with second-order information. Moreover, distinct from common adaptive schemes, we define the step size recursively as a function of the gradient norm and the prediction error in the optimistic update. We first analyze a variant where the step size requires knowledge of the Lipschitz constant of the Hessian. Under the additional assumption of Lipschitz continuous gradients, we further design a parameter-free version by tracking the Hessian Lipschitz constant locally and ensuring the iterates remain bounded. We also evaluate the practical performance of our algorithm by comparing it to existing second-order algorithms for minimax optimization.

## 1 Introduction

In this paper, we consider the min-max optimization problem, also known as the saddle point problem:

$$\min_{\mathbf{x} \in \mathbb{R}^m} \max_{\mathbf{y} \in \mathbb{R}^n} f(\mathbf{x}, \mathbf{y}), \tag{1}$$

where the objective function $f : \mathbb{R}^m \times \mathbb{R}^n \to \mathbb{R}$ is twice differentiable and convex-concave, i.e., $f(\cdot, \mathbf{y})$ is convex for any fixed $\mathbf{y} \in \mathbb{R}^n$ and $f(\mathbf{x}, \cdot)$ is concave for any fixed $\mathbf{x} \in \mathbb{R}^m$. The saddle point problem (1) is a fundamental formulation in machine learning and optimization and naturally emerges in several applications, including constrained and primal-dual optimization [1, 2], (multi-agent) games [3], reinforcement learning [4], and generative adversarial networks [5, 6]. The saddle point problem, which can be interpreted as a particular instance of variational inequalities and monotone inclusion problems [2], has a rich history dating back to [7]. We often solve (1) using iterative, first-order methods due to their simplicity and low per-iteration complexity. Over the past decades, various first-order algorithms have been proposed and analyzed for different settings [1, 8–15]. Under the assumption that the gradient of $f$ is Lipschitz, the aforementioned methods converge at a rate of $\mathcal{O}(1/T)$, where $T$ is the number of iterations. This rate is optimal for first-order methods [10, 16, 17].

Recently, there has been a surge of interest in higher-order methods for solving (1) [18–23], mirroring the trend in convex minimization literature [24–28]. In general, these methods exploit higher-order derivatives of $f$ to achieve faster convergence rates. From a practical viewpoint, any method

38th Conference on Neural Information Processing Systems (NeurIPS 2024).

involving third and higher-order derivatives is essentially a *conceptual* framework; it is unknown how to efficiently solve auxiliary problems involving higher-order derivatives, making it virtually impossible to efficiently implement methods beyond second-order [24]. Therefore, we focus on second-order methods and review the literature accordingly.

The existing literature on second-order algorithms for minimax optimization, capable of achieving the optimal convergence rate of $\mathcal{O}(1/T^{1.5})$, falls into two categories. The first group requires solving a linear system of equations (or matrix inversion) for their updates but needs a "line search" scheme to select the step size properly. This includes methods such as the Newton proximal extragradient method [18, 19], second-order extensions of the mirror-prox algorithm [20], and the second-order optimistic method [21]. These methods impose a cyclic and implicit relationship between the step size and the next iterate, necessitating line search mechanisms to compute a valid selection that meets the specified conditions.

The second group, which includes [22, 23], does not require a line search scheme and bypasses the implicit definitions and search subroutines. They follow the template of the cubic regularized Newton method [29] for convex minimization and solve an analogous "cubic variational inequality sub-problem" per iteration. Despite having explicit parameter definitions, these methods require specialized sub-solvers to obtain approximate solutions to the auxiliary problem, increasing the per-iteration complexity. Moreover, both groups of algorithms rely vitally on the precise knowledge of the objective's Hessian Lipschitz constant.

While the above frameworks achieve the optimal iteration complexity for second-order methods, their requirement for performing a line search or solving a cubic sub-problem limits their applicability. Recently, the authors in [30] proposed a method with optimal iteration complexity that requires neither the line search nor the solution of an auxiliary sub-problem. In each iteration, they compute a "candidate" next point $\mathbf{y}_t$ from the base point $\mathbf{x}_{t-1}$. However, unless the step size satisfies a "large step condition", which requires the exact knowledge of the Hessian's Lipschitz constant, the base point remains the same for the next iteration, slowing down the convergence in practice. Therefore, it remains an open problem to design a simple, efficient, and optimal second-order method without the need for line search, auxiliary sub-problems, and the knowledge of the Hessian's Lipschitz constant.

**Our Contributions.** Motivated by the aforementioned shortcomings in the literature, our proposed framework completely eliminates the need for line search and backtracking by providing a closed-form, explicit, simple iterate recursion with a data-adaptive step size that adjusts according to local information. In doing so, we develop a parameter-free method that does not require any problem parameters, such as the Lipschitz constant of the Hessian. The key to our simple, parameter-free algorithm is a careful combination of the second-order optimistic algorithm and adaptive regularization of the second-order update. We summarize the highlights of our work as follows:

1. We first present an adaptive second-order optimistic method that achieves the optimal rate of $\mathcal{O}(1/T^{1.5})$ without requiring any form of line search, assuming the Hessian is Lipschitz and its associated constant is known. We introduce a recursive, adaptive update rule for the step size as a function of the gradient and the Hessian at the current and previous iterations. Our step size satisfies a specific error condition, ensuring sufficient progress while growing at a favorable rate to establish optimal convergence rates.

2. Under the additional, mild assumption that the gradient is Lipschitz, we propose a *parameter-free* version with the same optimal rates which *adaptively* adjusts the regularization factor by means of a local curvature estimator. This method is completely oblivious to any problem-dependent parameter including Lipschitz constant(s) and the initialization. Importantly, we achieve this parameter-free guarantee without artificially imposing bounded iterates, which is a common yet restrictive assumption in the study of adaptive methods in minimization [31–33] and min-max [34, 35] literature.

## 2 Preliminaries

An optimal solution of (1) denoted by $(\mathbf{x}^*, \mathbf{y}^*)$ is called a *saddle point* of $f$, as it satisfies the property $f(\mathbf{x}^*, \mathbf{y}) \leq f(\mathbf{x}^*, \mathbf{y}^*) \leq f(\mathbf{x}, \mathbf{y}^*)$ for any $\mathbf{x} \in \mathbb{R}^m$, $\mathbf{y} \in \mathbb{R}^n$. Given this notion of optimality, one can measure the suboptimality of any $(\mathbf{x}, \mathbf{y})$ using the primal-dual gap, i.e., $\mathrm{Gap}(\mathbf{x}, \mathbf{y}) := \max_{\tilde{\mathbf{y}} \in \mathbb{R}^n} f(\mathbf{x}, \tilde{\mathbf{y}}) - \min_{\tilde{\mathbf{x}} \in \mathbb{R}^m} f(\tilde{\mathbf{x}}, \mathbf{y})$. However, it could be vacuous if not restricted to a bounded region. For instance, when $f(\mathbf{x}, \mathbf{y}) = \langle \mathbf{x}, \mathbf{y} \rangle$, this measure is always $\mathrm{Gap}(\mathbf{x}, \mathbf{y}) = +\infty$, except at

the saddle point $(0,0)$. To remedy this issue, we consider the restricted primal-dual gap function:

$$\text{Gap}_{\mathcal{X} \times \mathcal{Y}}(\mathbf{x}, \mathbf{y}) := \max_{\tilde{\mathbf{y}} \in \mathcal{Y}} f(\mathbf{x}, \tilde{\mathbf{y}}) - \min_{\tilde{\mathbf{x}} \in \mathcal{X}} f(\tilde{\mathbf{x}}, \mathbf{y}), \tag{Gap}$$

where $\mathcal{X} \subset \mathbb{R}^m$ and $\mathcal{Y} \subset \mathbb{R}^n$ are two compact sets containing the optimal solutions of problem (1). The restricted gap function is a valid merit function (see [1, 11]), and has been used as a measure of suboptimality for min-max optimization [1]. Next, we state our assumptions on Problem (1).

**Assumption 2.1.** *The objective $f$ is convex-concave, i.e., $f(\cdot, \mathbf{y})$ is convex for any fixed $\mathbf{y} \in \mathbb{R}^n$ and $f(\mathbf{x}, \cdot)$ is concave for any fixed $\mathbf{x} \in \mathbb{R}^m$.*

**Assumption 2.2.** *The Hessian of $f$ is $L_2$-Lipschitz, i.e., $\|\nabla^2 f(\mathbf{x}_1, \mathbf{y}_1) - \nabla^2 f(\mathbf{x}_2, \mathbf{y}_2)\| \leq L_2 \|(\mathbf{x}_1 - \mathbf{x}_2, \mathbf{y}_1 - \mathbf{y}_2)\|$ for any $(\mathbf{x}_1, \mathbf{y}_1), (\mathbf{x}_2, \mathbf{y}_2) \in \mathbb{R}^m \times \mathbb{R}^n$.*

Assumptions 2.1 and 2.2 are standard in the study of second-order methods in min-max optimization and constitute our core assumption set. That said, *only* for the parameter-free version of our proposed algorithm, we will require the additional condition that the gradient of $f$ is $L_1$-Lipschitz.

**Assumption 2.3.** *The gradient of $f$ is $L_1$-Lipschitz, i.e., $\|\nabla f(\mathbf{x}_1, \mathbf{y}_1) - \nabla f(\mathbf{x}_2, \mathbf{y}_2)\| \leq L_1 \|(\mathbf{x}_1 - \mathbf{x}_2, \mathbf{y}_1 - \mathbf{y}_2)\|$ for any $(\mathbf{x}_1, \mathbf{y}_1), (\mathbf{x}_2, \mathbf{y}_2) \in \mathbb{R}^m \times \mathbb{R}^n$.*

To simplify the notation, we define the concatenated vector of variables as $\mathbf{z} = (\mathbf{x}, \mathbf{y}) \in \mathbb{R}^m \times \mathbb{R}^n$, and define the operator $\mathbf{F} : \mathbb{R}^{m+n} \to \mathbb{R}^{m+n}$ at $\mathbf{z} = (\mathbf{x}, \mathbf{y})$ as

$$\mathbf{F}(\mathbf{z}) = [\nabla_{\mathbf{x}} f(\mathbf{x}, \mathbf{y}); -\nabla_{\mathbf{y}} f(\mathbf{x}, \mathbf{y})]. \tag{2}$$

Under Assumption 2.1, the operator $\mathbf{F}$ is *monotone*, i.e., $\langle \mathbf{F}(\mathbf{z}_1) - \mathbf{F}(\mathbf{z}_2), \mathbf{z}_1 - \mathbf{z}_2 \rangle \geq 0$ for any $\mathbf{z}_1, \mathbf{z}_2 \in \mathbb{R}^m \times \mathbb{R}^n$. Moreover, Assumption 2.2 implies that the Jacobian of $\mathbf{F}$, denoted by $\mathbf{F}'$, is $L_2$-Lipschitz, i.e., for any $\mathbf{z}_1, \mathbf{z}_2 \in \mathbb{R}^m \times \mathbb{R}^n$ we have $\|\mathbf{F}'(\mathbf{z}_1) - \mathbf{F}'(\mathbf{z}_2)\|_{op} \leq \frac{L_2}{2} \|\mathbf{z}_1 - \mathbf{z}_2\|$. This is referred to as *second-order* smoothness [20, 21]. Similarly, Assumption 2.3 implies that the operator $\mathbf{F}$ itself is $L_1$-Lipschitz, i.e., $\|\mathbf{F}(\mathbf{z}_1) - \mathbf{F}(\mathbf{z}_2)\| \leq L_1 \|\mathbf{z}_1 - \mathbf{z}_2\|$ for any $\mathbf{z}_1, \mathbf{z}_2 \in \mathbb{R}^m \times \mathbb{R}^n$.

Finally, the following classic lemma plays a key role in our convergence analysis, as it provides an upper bound on the restricted primal-dual gap at the averaged iterate. Proof can be found in [36].

**Lemma 2.1.** *Suppose Assumption 2.1 holds. Consider $\theta_1, \ldots, \theta_T \geq 0$ with $\sum_{t=1}^{T} \theta_t = 1$ and $\mathbf{z}_1 = (\mathbf{x}_1, \mathbf{y}_1), \ldots, \mathbf{z}_T = (\mathbf{x}_T, \mathbf{y}_T) \in \mathbb{R}^m \times \mathbb{R}^n$. Define the average iterates as $\bar{\mathbf{x}}_T = \sum_{t=1}^{T} \theta_t \mathbf{x}_t$ and $\bar{\mathbf{y}}_T = \sum_{t=1}^{T} \theta_t \mathbf{y}_t$. Then, $f(\bar{\mathbf{x}}_T, \mathbf{y}) - f(\mathbf{x}, \bar{\mathbf{y}}_T) \leq \sum_{t=1}^{T} \theta_t \langle \mathbf{F}(\mathbf{z}_t), \mathbf{z}_t - \mathbf{z} \rangle$ for any $(\mathbf{x}, \mathbf{y}) \in \mathbb{R}^m \times \mathbb{R}^n$.*

For simplicity and ease of delivery, our algorithm and analysis are based on the operator representation of Problem (1). By means of Lemma 2.1, our derivations with respect to the operator $\mathbf{F}$ imply convergence in terms of the (restricted) primal-dual (Gap) function.

## 3 Background on optimistic methods

At its core, our algorithm is a second-order variant of the optimistic scheme for solving min-max problems [12, 14, 15, 21]. As discussed in [36, 37], the optimistic framework can be considered as an approximation of the proximal point method (PPM) [38, 39], which is given by $\mathbf{z}_{t+1} = \mathbf{z}_t - \eta_t \mathbf{F}(\mathbf{z}_{t+1})$. To highlight this connection, note that PPM is an implicit method since the operator $\mathbf{F}$ is evaluated at the *next* iterate $\mathbf{z}_{t+1}$. The first-order optimistic method approximates PPM by a careful combination of gradients in two consecutive iterates. The second-order variant [21], however, jointly uses first and second-order information, which we describe next. Its key idea is to approximate the "implicit gradient" $\mathbf{F}(\mathbf{z}_{t+1})$ in PPM by its linear approximation $\mathbf{F}(\mathbf{z}_t) + \mathbf{F}'(\mathbf{z}_t)(\mathbf{z} - \mathbf{z}_t)$ around the current point $\mathbf{z}_t$, and to correct this "prediction" with the error associated with the previous iteration. Specifically, the correction term, denoted by $\mathbf{e}_t := \mathbf{F}(\mathbf{z}_t) - \mathbf{F}(\mathbf{z}_{t-1}) - \mathbf{F}'(\mathbf{z}_{t-1})(\mathbf{z}_t - \mathbf{z}_{t-1})$, is the difference between $\mathbf{F}(\mathbf{z}_t)$ and its prediction at $t-1$. To express in a formal way,

$$\eta_t \mathbf{F}(\mathbf{z}_{t+1}) \approx \underbrace{\eta_t [\mathbf{F}(\mathbf{z}_t) + \mathbf{F}'(\mathbf{z}_t)(\mathbf{z}_{t+1} - \mathbf{z}_t)]}_{\text{prediction term}} + \underbrace{\eta_{t-1} [\mathbf{F}(\mathbf{z}_t) - \mathbf{F}(\mathbf{z}_{t-1}) - \mathbf{F}'(\mathbf{z}_{t-1})(\mathbf{z}_t - \mathbf{z}_{t-1})]}_{\text{correction term}}. \tag{3}$$

The rationale behind the optimism is that if the prediction errors in two consecutive rounds do not vary much, i.e., $\eta_t \mathbf{e}_{t+1} \approx \eta_{t-1} \mathbf{e}_t$, then the correction term should help reduce the approximation

---

**Algorithm 1** Adaptive Second-order Optimistic Method

---

1: **Input**: Initial points $\mathbf{z}_0 = \mathbf{z}_1 \in \mathbb{R}^m \times \mathbb{R}^n$, initial parameters $\eta_0 = 0$ and $\lambda_0 > 0$
2: **for** $t = 1, \ldots, T$ **do**
3:     Set: $\mathbf{e}_t = \mathbf{F}(\mathbf{z}_t) - \mathbf{F}(\mathbf{z}_{t-1}) - \mathbf{F}'(\mathbf{z}_{t-1})(\mathbf{z}_t - \mathbf{z}_{t-1})$
4:     Set the step size parameters

$$\lambda_t = \begin{cases} L_2 & \textbf{(I)} \\ \max\left\{\lambda_{t-1}, \frac{2\|\mathbf{e}_t\|}{\|\mathbf{z}_t - \mathbf{z}_{t-1}\|^2}\right\} & \textbf{(II)} \end{cases} \qquad \eta_t = \frac{\lambda_t}{2\left(\eta_{t-1}\|\mathbf{e}_t\| + \sqrt{\eta_{t-1}^2\|\mathbf{e}_t\|^2 + \lambda_t\|\mathbf{F}(\mathbf{z}_t)\|}\right)}$$

5:     Update: $\mathbf{z}_{t+1} = \mathbf{z}_t - (\lambda_t\mathbf{I} + \eta_t\mathbf{F}'(\mathbf{z}_t))^{-1}(\eta_t\mathbf{F}(\mathbf{z}_t) + \eta_{t-1}\mathbf{e}_t)$
6: **end for**
7: **return** $\bar{\mathbf{z}}_{T+1} = (\sum_{t=0}^T \eta_t)^{-1}\sum_{t=0}^T \eta_t\mathbf{z}_{t+1}$

---

error and thus lead to a faster convergence rate. Replacing $\eta_t\mathbf{F}(\mathbf{z}_{t+1})$ by its approximation in (3) and rearranging the terms leads to the update rule of the second-order optimistic method:

$$\mathbf{z}_{t+1} = \mathbf{z}_t - (\mathbf{I} + \eta_t\mathbf{F}'(\mathbf{z}_t))^{-1}(\eta_t\mathbf{F}(\mathbf{z}_t) + \eta_{t-1}\mathbf{e}_t). \tag{4}$$

The key challenge is to control the discrepancy between the second-order optimistic method and PPM. This is equivalent to managing the deviation between the updates of the second-order optimistic method and the PPM update. We achieve this by checking an additional condition denoted by

$$\eta_t\|\mathbf{e}_{t+1}\| := \eta_t\|\mathbf{F}(\mathbf{z}_{t+1}) - \mathbf{F}(\mathbf{z}_t) - \mathbf{F}'(\mathbf{z}_t)(\mathbf{z}_{t+1} - \mathbf{z}_t)\| \leq \alpha\|\mathbf{z}_{t+1} - \mathbf{z}_t\|, \tag{5}$$

where $\alpha \in (0, 0.5)$. Note that if the prediction term perfectly predicts the prox step, we recover the PPM update and the condition holds with $\alpha = 0$. For the standard second-order optimistic algorithm in (4), we need to select $\alpha \leq 0.5$. The condition in (5) emerges solely from the convergence analysis.

While the above method successfully achieves the optimal complexity of $\mathcal{O}(1/T^{1.5})$, there remains a major challenge in selecting $\eta_t$. A naïve choice guided by the condition in (5) results in an *implicit* parameter update. Specifically, note that the error condition in (5) involves both $\eta_t$ and the next iterate $\mathbf{z}_{t+1}$, but $\mathbf{z}_{t+1}$ is computed only *after* the step size $\eta_t$ is determined. Consequently, we can test whether the condition in (5) is satisfied only after selecting the step size $\eta_t$. The authors in [21] tackled this challenge with a direct approach and proposed a "line search scheme", where $\eta_k$ is backtracked until (5) is satisfied. While their line search scheme requires only a constant number of backtracking steps on average, it is desirable to design simpler line search-free algorithms for practical and efficiency purposes.

## 4 Proposed algorithms

As discussed, the current theory of second-order optimistic methods requires line search due to the implicit structure of (5). In this section, we address this issue and present a class of second-order methods that, without any line search scheme, are capable of achieving the optimal complexity for convex-concave min-max setting. To begin, we first present a general version of the second-order optimistic method by introducing an additional scaling parameter $\lambda_t$. Specifically, the update is

$$\mathbf{z}_{t+1} = \mathbf{z}_t - (\lambda_t\mathbf{I} + \eta_t\mathbf{F}'(\mathbf{z}_t))^{-1}(\eta_t\mathbf{F}(\mathbf{z}_t) + \eta_{t-1}\mathbf{e}_t). \tag{6}$$

when $\lambda_t = 1$, we recover the update in (4). Crucially, the regularization factor $\lambda_t$ enables flexibility in choosing the parameters of our proposed algorithm and plays a vital role in achieving the parameter-free design, which does not need the knowledge of the Lipschitz constant. What remains to be shown is the update rule for $\eta_t$ and $\lambda_t$. In the following sections, we present two adaptive update policies for these parameters. The first policy is line-search-free, explicit, and only requires knowledge of $L_2$. The second approach does not require knowledge of $L_2$ and is completely parameter-free, but it requires an additional assumption that $\mathbf{F}$ is $L_1$-Lipschitz, which is satisfied when $\nabla f$ is $L_1$-Lipschitz (see Assumption 2.3).

**Adaptive and line search-free second-order optimistic method (Option I).** In our first proposed method, we set the parameter $\lambda_t$ to be a fixed value $\lambda$ and update the parameter $\eta_t$ using the policy:

$$\eta_t = \frac{4\alpha\lambda^2}{\eta_{t-1}L_2\|\mathbf{e}_t\| + \sqrt{(\eta_{t-1}L_2\|\mathbf{e}_t\|)^2 + 8\alpha\lambda^2 L_2\|\mathbf{F}(\mathbf{z}_t)\|}}. \tag{7}$$

As we observe, $\eta_t$ only depends on the information that is available at time $t$, including the error term norm $\|\mathbf{e}_t\|$ and the operator norm $\|\mathbf{F}(\mathbf{z}_t)\|$. Hence, the update is explicit and does not require any form of backtracking or line search. That said, it requires the knowledge of the Lipschitz constant of the Jacobian $\mathbf{F}'$ denoted by $L_2$. We should note that $\lambda > 0$ in this case is a free parameter, and we set it as $\lambda = L_2$ to be consistent with the parameter-free method in the next section. The update for $\eta_t$ might seem counter-intuitive at first glance, but as we elaborate upon its derivation in the next section, it is fully justified by optimizing the upper bounds corresponding to the optimistic method.

**Parameter-free adaptive second-order optimistic method (Option II).** While the expression for step size $\eta_t$ in (7) is explicit and adaptive to the optimization process, however, it depends on the Hessian's Lipschitz constant $L_2$. Next, we discuss how to make the method parameter-free, so that the algorithm parameters $\lambda_t$ and $\eta_t$ do not depend on the smoothness constant(s) or any problem-dependent parameters. Specifically, we propose the following update for $\lambda_t$ and $\eta_t$:

$$\eta_t = \frac{2\alpha\lambda_t}{\eta_{t-1}\|\mathbf{e}_t\| + \sqrt{\eta_{t-1}^2\|\mathbf{e}_t\|^2 + 4\alpha\lambda_t\|\mathbf{F}(\mathbf{z}_t)\|}}, \quad \text{where } \lambda_t = \max\left\{\lambda_{t-1}, \frac{2\|\mathbf{e}_t\|}{\|\mathbf{z}_{t-1} - \mathbf{z}_t\|^2}\right\}. \quad (8)$$

These updates are explicit, adaptive, and parameter-free. In the next section, we justify these updates.

## 5  Main ideas behind the suggested updates

Before we delve into the convergence theorems, we proceed by explaining the particular choice of algorithm parameters and the derivation process behind their design, through which we will motivate how we eliminate the need for iterative line search.

**Rationale behind the update of Option I.** First, we motivate the design process for updating $\eta_t$ and $\lambda$ in Option (**I**), guided by the convergence analysis. We illustrate the technical details leading to the parameter choices in Step 4 by introducing a template equality that forms the basis of our analysis.

**Proposition 5.1.** *Let $\{\mathbf{z}_t\}_{t=0}^{T+1}$ be generated by Algorithm 1. Define the "approximation error" as* $\mathbf{e}_{t+1} \triangleq \mathbf{F}(\mathbf{z}_{t+1}) - \mathbf{F}(\mathbf{z}_t) - \mathbf{F}'(\mathbf{z}_t)(\mathbf{z}_{t+1} - \mathbf{z}_t)$. *Then for any $\mathbf{z} \in \mathbb{R}^d$, we have*

$$\sum_{t=1}^{T} \eta_t\langle\mathbf{F}(\mathbf{z}_{t+1}), \mathbf{z}_{t+1} - \mathbf{z}\rangle = \sum_{t=1}^{T} \frac{\lambda_t}{2}\left(\|\mathbf{z}_t - \mathbf{z}\|^2 - \|\mathbf{z}_{t+1} - \mathbf{z}\|^2\right) - \sum_{t=1}^{T} \frac{\lambda_t}{2}\|\mathbf{z}_t - \mathbf{z}_{t+1}\|^2$$
$$+ \underbrace{\eta_T\langle\mathbf{e}_{T+1}, \mathbf{z}_{T+1} - \mathbf{z}\rangle}_{(A)} + \sum_{t=1}^{T} \underbrace{\eta_{t-1}\langle\mathbf{e}_t, \mathbf{z}_t - \mathbf{z}_{t+1}\rangle}_{(B)}. \quad (9)$$

As we observe in the above bound, if we set $\lambda_t$ to be constant ($\lambda_t = \lambda$), then the first summation term on the right-hand side will telescope. On top of that, if we apply the Cauchy-Schwarz inequality and Young's inequality on terms (A) and (B) and regroup the matching expressions, we would obtain $\sum_{t=1}^{T} \eta_t\langle\mathbf{F}(\mathbf{z}_{t+1}), \mathbf{z}_{t+1}-\mathbf{z}\rangle \le \frac{\lambda}{2}\|\mathbf{z}_1-\mathbf{z}\|^2 - \frac{\lambda}{4}\|\mathbf{z}_{T+1}-\mathbf{z}\|^2 + \sum_{t=1}^{T}\left(\frac{\eta_t^2}{\lambda}\|\mathbf{e}_{t+1}\|^2 - \frac{\lambda}{4}\|\mathbf{z}_t-\mathbf{z}_{t+1}\|^2\right)$. We make two remarks regarding the inequality above. (*i*) By using Lemma 2.1 with $\theta_t = \frac{\eta_t}{\sum_{t=1}^{T}\eta_t}$ for $1 \le t \le T$, the left-hand side can be lower bounded by $\left(\sum_{t=1}^{T}\eta_t\right)(f(\bar{\mathbf{x}}_{T+1}, \mathbf{y}) - f(\mathbf{x}, \bar{\mathbf{y}}_{T+1}))$, where the averaged iterate $\bar{\mathbf{z}}_{T+1} = (\bar{\mathbf{x}}_{T+1}, \bar{\mathbf{y}}_{T+1})$ is given by $\bar{\mathbf{z}}_{T+1} = \frac{1}{\sum_{t=1}^{T}\eta_t}\sum_{t=1}^{T}\eta_t\mathbf{z}_{t+1}$. (*ii*) If we can show that the summation on the right-hand side is non-positive and divide both sides by $\sum_{t=1}^{T}\eta_t$, we obtain a convergence rate of $\mathcal{O}(1/\sum_{t=1}^{T}\eta_t)$ for (Gap) at the averaged iterate.

To obtain the optimal rate of $\mathcal{O}(1/T^{1.5})$, the analysis guides us to be more conservative with the latter point and ensure that the summation on the right-hand side is strictly negative (see Section 6 for further details). Specifically, we require each error term in the summation to satisfy $\frac{\eta_t^2}{\lambda}\|\mathbf{e}_{t+1}\|^2 - \frac{\lambda}{4}\|\mathbf{z}_t-\mathbf{z}_{t+1}\|^2 \le -\left(\frac{1}{4} - \alpha^2\right)\lambda\|\mathbf{z}_t-\mathbf{z}_{t+1}\|^2$ for a given $\alpha \in (0, \frac{1}{2})$. Rearranging the expressions we obtain $\eta_t^2\|\mathbf{e}_{t+1}\|^2 \le \alpha^2\lambda^2\|\mathbf{z}_t-\mathbf{z}_{t+1}\|^2$, and we retrieve an analog of the error condition (5) by simply taking the square root of both sides. A naïve approach would be to choose $\eta_t$ small enough to satisfy the condition. However, since our convergence rate is of the form $\sum_{t=1}^{T}\eta_t$, this approach would also slow down the convergence of our algorithm and achieve a sub-optimal rate.

Hence, our goal is to *select the largest possible $\eta_t$ that satisfies the condition in* (5). Next, we will explain how we come up with an explicit update rule for step size $\eta_t$ that achieves this goal. Our strategy is quite simple; we first rewrite the inequality of interest as

$$\frac{\eta_t \|\mathbf{e}_{t+1}\|}{\alpha\lambda\|\mathbf{z}_t - \mathbf{z}_{t+1}\|} \leq 1. \tag{10}$$

Then, we derive an upper bound for the term on the left-hand side that depends only on quantities available at iteration $t$. A sufficient condition for (10) would be showing that the upper bound of $\frac{\eta_t\|\mathbf{e}_{t+1}\|}{\alpha\lambda\|\mathbf{z}_t - \mathbf{z}_{t+1}\|}$ is less than 1. Note that by Assumption 2.2, we can upper bound $\|\mathbf{e}_{t+1}\|$ and write $\frac{\eta_t\|\mathbf{e}_{t+1}\|}{\alpha\lambda\|\mathbf{z}_t - \mathbf{z}_{t+1}\|} \leq \frac{\eta_t L_2\|\mathbf{z}_t - \mathbf{z}_{t+1}\|^2}{2\alpha\lambda\|\mathbf{z}_t - \mathbf{z}_{t+1}\|} = \frac{\eta_t L_2\|\mathbf{z}_t - \mathbf{z}_{t+1}\|}{2\alpha\lambda}$. As the final component, we derive an upper bound for $\|\mathbf{z}_t - \mathbf{z}_{t+1}\|$ that only depends on the information available at time $t$. In the next lemma, which follows from the update rule and the fact that $\mathbf{F}$ is monotone, we accomplish this goal. The proof is in Appendix A.2.

**Lemma 5.2.** *Suppose that Assumption 2.1 holds. Then, the update rule in Step 5 in Algorithm 1 implies* $\|\mathbf{z}_t - \mathbf{z}_{t+1}\| \leq \frac{1}{\lambda_t}\eta_t\|\mathbf{F}(\mathbf{z}_t)\| + \frac{1}{\lambda_t}\eta_{t-1}\|\mathbf{e}_t\|$.

We combine Lemma 5.2 for $\lambda_t = \lambda$ with the previous expression and rearrange the terms to obtain

$$\frac{\eta_t L_2\|\mathbf{z}_t - \mathbf{z}_{t+1}\|}{2\alpha\lambda} \leq \frac{\eta_t L_2(\eta_t\|\mathbf{F}(\mathbf{z}_t)\| + \eta_{t-1}\|\mathbf{e}_t\|)}{2\alpha\lambda^2}. \tag{11}$$

Hence, we obtained an *explicit* upper bound for the left hand side of (10) that only depends on terms at iteration $t$ or before. Therefore, a sufficient condition for satisfying (10) is ensuring that $\frac{\eta_t L_2(\eta_t\|\mathbf{F}(\mathbf{z}_t)\| + \eta_{t-1}\|\mathbf{e}_t\|)}{2\alpha\lambda^2} \leq 1$. Since we aim for the largest possible choice of $\eta_t$, we intend to satisfy this condition with equality. After rearranging, we end up with the following expression:

$$\eta_t(\eta_t\|\mathbf{F}(\mathbf{z}_t)\| + \eta_{t-1}\|\mathbf{e}_t\|) = \frac{2\alpha\lambda^2}{L_2}. \tag{12}$$

The expression in (12) is a quadratic equation in $\eta_t$ and it is an *explicit* expression where all the terms are available at the beginning of iteration $t$. Solving for $\eta_t$ leads to the expression in (7).

**Rationale behind the update of Option II.** Choosing the regularization parameter $\lambda_t$ properly is the key piece of the puzzle. First, recall the error term $\sum_{t=1}^{T} \frac{\lambda_t}{2}\left(\|\mathbf{z}_t - \mathbf{z}\|^2 - \|\mathbf{z}_{t+1} - \mathbf{z}\|^2\right)$ from Proposition 5.1. When $\lambda_t$ is time-varying, this summation no longer telescopes. A standard technique in adaptive gradient methods to resolve this issue (see, e.g., [40, Theorem 2.13]) involves selecting $\lambda_t$ to be monotonically non-decreasing and showing that the iterates $\{\mathbf{z}_t\}_{t\geq0}$ are bounded. We follow this approach, and in the next proposition, we investigate the possibility of ensuring that the distance of the iterates to the optimal solution, $\|\mathbf{z}_t - \mathbf{z}^*\|^2$, remains bounded.

**Proposition 5.3.** *Let $\{\mathbf{z}_t\}_{t=0}^{T+1}$ be generated by Algorithm 1 and $\mathbf{z}^* \in \mathbb{R}^m \times \mathbb{R}^n$ be a solution to Problem* (1). *Then,*

$$\frac{1}{2}\|\mathbf{z}_{T+1} - \mathbf{z}^*\|^2 \leq \frac{1}{2}\|\mathbf{z}_1 - \mathbf{z}^*\|^2 - \sum_{t=1}^{T}\frac{1}{2}\|\mathbf{z}_t - \mathbf{z}_{t+1}\|^2 + \sum_{t=1}^{T}\underbrace{\frac{\eta_{t-1}}{\lambda_t}\langle\mathbf{e}_t, \mathbf{z}_t - \mathbf{z}_{t+1}\rangle}_{(A)}$$

$$+ \underbrace{\frac{\eta_T}{\lambda_T}\langle\mathbf{e}_{T+1}, \mathbf{z}_{T+1} - \mathbf{z}^*\rangle}_{(B)} + \sum_{t=2}^{T}\underbrace{\left(\frac{1}{\lambda_{t-1}} - \frac{1}{\lambda_t}\right)\eta_{t-1}\langle\mathbf{e}_t, \mathbf{z}_t - \mathbf{z}^*\rangle}_{(C)}. \tag{13}$$

To derive the boundedness of $\{\mathbf{z}_t\}_{t\geq1}$ from (13), all error terms (A), (B), and (C) in (13) should be upper bounded. As detailed in the proof of Lemma C.1, we can apply Assumption 2.3 to control the second term (B) and it does not impose restrictions on our choice of $\eta_t$ and $\lambda_t$. To control term (A) and (C), we apply Cauchy-Schwarz and Young's inequalities individually; we get $\frac{\eta_{t-1}}{\lambda_t}\langle\mathbf{e}_t, \mathbf{z}_t - \mathbf{z}_{t+1}\rangle \leq \frac{\eta_{t-1}^2}{\lambda_t^2}\|\mathbf{e}_t\|^2 + \frac{1}{4}\|\mathbf{z}_t - \mathbf{z}_{t+1}\|^2$, and also $(\frac{1}{\lambda_{t-1}} - \frac{1}{\lambda_t})\eta_{t-1}\langle\mathbf{e}_t, \mathbf{z}_t - \mathbf{z}^*\rangle \leq \frac{\eta_{t-1}^2}{\lambda_t^2}\|\mathbf{e}_t\|^2 + \frac{1}{4}(\frac{\lambda_t}{\lambda_{t-1}} - 1)^2\|\mathbf{z}_t - \mathbf{z}^*\|^2$, respectively. Combining the new terms obtained from (A) and (C) and summing from $t = 1$ to $T$, we obtain $\sum_{t=1}^{T}\frac{2\eta_t^2}{\lambda_{t+1}^2}\|\mathbf{e}_{t+1}\|^2 + \frac{1}{4}\sum_{t=1}^{T}\|\mathbf{z}_t - \mathbf{z}_{t+1}\|^2 + \sum_{t=1}^{T}\frac{1}{4}(\frac{\lambda_t}{\lambda_{t-1}} - 1)^2\|\mathbf{z}_t - \mathbf{z}^*\|^2$. The

last term will remain in the recursive formula, hence manageable. On the other hand, we need to make sure that the first two terms can be canceled out by the negative terms we have in (13). Thus, we need to enforce the condition $\frac{2\eta_t^2}{\lambda_{t+1}^2}\|\mathbf{e}_{t+1}\|^2 + \frac{1}{4}\|\mathbf{z}_t - \mathbf{z}_{t+1}\|^2 - \frac{1}{2}\|\mathbf{z}_t - \mathbf{z}_{t+1}\|^2 \leq -\left(\frac{1}{4} - 2\alpha^2\right)\|\mathbf{z}_t - \mathbf{z}_{t+1}\|^2$, where $\alpha \in (0, \frac{1}{2\sqrt{2}})$. This condition can be simplified as

$$\frac{\eta_t^2\|\mathbf{e}_{t+1}\|^2}{\lambda_{t+1}^2\|\mathbf{z}_t - \mathbf{z}_{t+1}\|^2} \leq \alpha^2 \quad \Leftrightarrow \quad \frac{\eta_t\|\mathbf{e}_{t+1}\|}{\alpha\lambda_{t+1}\|\mathbf{z}_t - \mathbf{z}_{t+1}\|} \leq 1. \tag{14}$$

Comparing with (11), we observe that the difference is that $\lambda$ is replaced by $\lambda_{t+1}$. Thus, we propose to follow a similar update rule for $\eta_t$ as in (7). However, recall that $L_2$ appears in the update rule of (7), yet we do not have the knowledge of $L_2$ in this setting. Hence, we assume that we can compute a sequence of Lipschitz constant estimates $\{\hat{L}_2^{(t)}\}$ at each iteration $t$. The construction of such Lipschitz estimates will be evident later from our analysis. Specifically, in the update rule of (8), we will replace $\lambda$ by $\lambda_t$ and replace $L_2$ by $\hat{L}_2^{(t)}$, leading to the expression

$$\eta_t = \frac{4\alpha\lambda_t^2}{\eta_{t-1}\|\mathbf{e}_t\| + \sqrt{(\eta_{t-1}\hat{L}_2^{(t)}\|\mathbf{e}_t\|)^2 + 8\alpha\lambda_t^2\hat{L}_2^{(t)}\|\mathbf{F}(\mathbf{z}_t)\|}}. \tag{15}$$

By relying on Lemma 5.2 and following similar arguments, we can show that

$$\frac{\eta_t\|\mathbf{e}_{t+1}\|}{\alpha\lambda_{t+1}\|\mathbf{z}_t - \mathbf{z}_{t+1}\|} \leq \frac{\lambda_t}{\lambda_{t+1}}\frac{L_2^{(t+1)}}{\hat{L}_2^{(t)}}, \tag{16}$$

where $L_2^{(t+1)} = \frac{2\|\mathbf{e}_{t+1}\|}{\|\mathbf{z}_{t+1} - \mathbf{z}_t\|^2}$ can be regarded as a "local" estimate of the Hessian's Lipschitz constant. Thus, to satisfy the condition in (14), the natural strategy would be to set $\lambda_t = \hat{L}_2^{(t)}$ and ensure that $L_2^{(t+1)} \leq \hat{L}_2^{(t+1)} = \lambda_{t+1}$. Finally, recall that the sequence $\{\lambda_t\}$ should be monotonically non-decreasing, i.e., $\lambda_{t+1} \geq \lambda_t$ for $t \in [T]$, leading to our update rule for $\lambda_{t+1}$ as shown in (8). This way, the right-hand side of (16) becomes $\frac{L_2^{(t+1)}}{\hat{L}_2^{(t+1)}} \leq 1$ and thus the error condition (14) is satisfied. By replacing $\hat{L}_2^{(t)}$ with $\lambda_t$ and simplifying the expression, we arrive at the update rule for $\eta_t$ in (8).

# 6 Convergence analysis

In this section, we present our convergence analysis for different variants of Algorithm 1. We first present the final convergence result for Option **(I)** of our proposed method. Besides the convergence bound in terms of (Gap), we provide a complementary convergence bound with respect to the norm of the operator, evaluated at the "best" iterate.

**Theorem 6.1.** *Suppose Assumptions 2.1 and 2.2 hold and let $\{\mathbf{z}_t\}_{t=0}^{T+1}$ be generated by Algorithm 1, where $\lambda_t = L_2$ (Option **(I)**) and $\alpha = 0.25$. Then $\|\mathbf{z}_t - \mathbf{z}^*\| \leq \frac{2}{\sqrt{3}}\|\mathbf{z}_1 - \mathbf{z}^*\|$ for all $t \geq 1$. Moreover,*

$$\text{Gap}_{\mathcal{X}\times\mathcal{Y}}(\bar{\mathbf{z}}_{T+1}) \leq \frac{\sup_{\mathbf{z}\in\mathcal{X}\times\mathcal{Y}}\|\mathbf{z}_1 - \mathbf{z}\|^2 \sqrt{2L_2\|\mathbf{F}(\mathbf{z}_1)\| + 36.25L_2^2\|\mathbf{z}_0 - \mathbf{z}^*\|^2}}{T^{1.5}}, \tag{17}$$

$$\min_{t\in\{2,\dots,T+1\}}\|\mathbf{F}(\mathbf{z}_t)\| \leq \frac{6\|\mathbf{z}_1 - \mathbf{z}^*\|\sqrt{16L_2\|\mathbf{F}(\mathbf{z}_1)\| + 290L_2^2\|\mathbf{z}_1 - \mathbf{z}^*\|^2}}{T}. \tag{18}$$

Theorem 6.1 guarantees that the iterates $\{\mathbf{z}_t\}_{t\geq0}$ always stay in a compact set $\{\mathbf{z} \in \mathbb{R}^d : \|\mathbf{z} - \mathbf{z}^*\| \leq \frac{2}{\sqrt{3}}\|\mathbf{z}_1 - \mathbf{z}^*\|\}$. Moreover, it demonstrates that the gap function at the weighted averaged iterate $\bar{\mathbf{z}}_{T+1}$ converges at the rate of $\mathcal{O}\left(T^{-1.5}\right)$, which is optimal and matches the lower bound in [23]. Finally, the convergence rate in (18) in terms of the operator norm also matches the state-of-the-art rate achieved by second-order methods [18], [41, Theorem 3.7], [30, Theorem 4.9 (a)].

*Proof Sketch of Theorem 6.1.* We begin with the convergence with respect to (Gap) in (17). The proof consists of the following steps.

**Step 1:** As mentioned in Section 5, the choice of $\eta_t$ in (7) guarantees that $\eta_t \|\mathbf{e}_{t+1}\| \leq \alpha\lambda\|\mathbf{z}_t - \mathbf{z}_{t+1}\|$. This allows us to prove that the right-hand side of (9) is bounded by $\frac{\lambda}{2}\|\mathbf{z}_1 - \mathbf{z}\|^2 = \frac{L_2}{2}\|\mathbf{z}_1 - \mathbf{z}\|^2$. Hence, using Lemma 2.1, we have $\text{Gap}_{\mathcal{X}\times\mathcal{Y}}(\bar{\mathbf{z}}_{T+1}) \leq \sup_{\mathbf{z}\in\mathcal{X}\times\mathcal{Y}} \frac{L_2}{2}\|\mathbf{z}_1 - \mathbf{z}\|^2 (\sum_{t=1}^{T}\eta_t)^{-1}$.

**Step 2:** Next, our goal is to lower bound $\sum_{t=1}^{T}\eta_t$. By using the expression of $\eta_t$ in (7) we can show a lower bound on $\eta_t$ in terms of $\|\mathbf{e}_t\|$ and $\|\mathbf{F}(\mathbf{z}_t)\|$ as (formalized in Lemma B.2)

$$\eta_t \geq 2\alpha\lambda \left(\eta_{t-1}^2\|\mathbf{e}_t\|^2 + 2\alpha\lambda\|\mathbf{F}(\mathbf{z}_t)\|\right)^{-\frac{1}{2}} \geq \left(\frac{1}{4}\|\mathbf{z}_t - \mathbf{z}_{t-1}\|^2 + \frac{1}{\alpha\lambda}\|\mathbf{F}(\mathbf{z}_t)\|\right)^{-\frac{1}{2}}. \quad (19)$$

Additionally, we need to establish an upper bound on $\|\mathbf{F}(\mathbf{z}_t)\|$. By leveraging the update rule in (4) and Assumption 2.2, we show that $\|\mathbf{F}(\mathbf{z}_t)\| \leq \frac{(1+\alpha)\lambda}{\eta_{t-1}}\|\mathbf{z}_t - \mathbf{z}_{t-1}\| + \frac{\alpha\lambda}{\eta_{t-1}}\|\mathbf{z}_{t-1} - \mathbf{z}_{t-2}\|$ for $t \geq 2$ in Lemma B.2. Thus, $\|\mathbf{F}(\mathbf{z}_t)\|$ can be bounded in terms of $\|\mathbf{z}_t - \mathbf{z}_{t-1}\|$, $\|\mathbf{z}_{t-1} - \mathbf{z}_{t-2}\|$ and $\eta_{t-1}$. In addition, by using Proposition 5.3, we can establish that $\sum_{t=1}^{T}\|\mathbf{z}_{t+1} - \mathbf{z}_t\|^2 = \mathcal{O}(\|\mathbf{z}_1 - \mathbf{z}^*\|^2)$.

**Step 3:** By combining the ingredients above, with some algebraic manipulations we can show that $\sum_{t=1}^{T}\frac{1}{\eta_t^2} = \mathcal{O}\left(\|\mathbf{z}_1 - \mathbf{z}^*\|^2 + \frac{1}{\lambda}\|\mathbf{F}(\mathbf{z}_1)\|\right)$ (check Lemma B.3). Hence, using Hölder's inequality, it holds that $\sum_{t=0}^{T}\eta_t \geq T^{1.5}(\sum_{t=0}^{T}(1/\eta_t^2))^{-1/2}$. This finishes the proof for (17).

Finally, we prove the convergence rate with respect to the operator norm in (18). Essentially, we reuse the results we have established previously. By using the upper bounds on $\|\mathbf{F}(\mathbf{z})\|$ and $\sum_{t=1}^{T}\frac{1}{\eta_t}$ from Lemmas B.2 and B.3 respectively, and combining them with the bound $\sum_{t=1}^{T}\|\mathbf{z}_{t+1} - \mathbf{z}_t\|^2 = \mathcal{O}(\|\mathbf{z}_1 - \mathbf{z}^*\|^2)$ (see Proposition B.1), we show that $\sum_{t=2}^{T+1}\|\mathbf{F}(\mathbf{z}_t)\| = \mathcal{O}\left(\lambda\|\mathbf{z}_1 - \mathbf{z}^*\|\sqrt{\sum_{t=1}^{T}\frac{1}{\eta_t^2}}\right) = \mathcal{O}\left(\|\mathbf{z}_1 - \mathbf{z}^*\|\sqrt{\lambda^2\|\mathbf{z}_1 - \mathbf{z}^*\|^2 + \lambda\|\mathbf{F}(\mathbf{z}_1)\|}\right)$. Then the bound follows from the simple fact that $\min_{\{2,\ldots,T+1\}}\|\mathbf{F}(\mathbf{z}_t)\| \leq \frac{1}{T}\sum_{t=2}^{T+1}\|\mathbf{F}(\mathbf{z}_t)\|$. □

Next, we proceed to present the convergence results for Option **(II)** of our proposed method that is parameter-free. Note that if the initial scaling parameter $\lambda_1$ overestimates the Lipschitz constant $L_2$, we have $\lambda_t = \lambda_1$ for all $t \geq 1$. This is because we have $\frac{2\|\mathbf{e}_t\|}{\|\mathbf{z}_t - \mathbf{z}_{t-1}\|} \leq L_2$ by Assumption 2.2, and thus in this case the maximum in (8) will be always $\lambda_{t-1}$. As a result, $\lambda_t$ stays constant and the convergence analysis for Option **(I)** also applies here. Given this argument, in the following, we focus on the case where the initial scaling parameter $\lambda_1$ underestimates $L_2$, i.e., $\lambda_1 < L_2$. Moreover, it is rather trivial to establish that $\lambda_t < L_2$ for all $t \geq 1$ using induction.

**Theorem 6.2.** *Suppose Assumptions 2.1, 2.2, and 2.3 hold and let $\{\mathbf{z}_t\}_{t=0}^{T+1}$ be generated by Algorithm 1, where $\lambda_t$ is given by (8) (Option (II)) and $\alpha = 0.25$. Assume that $\lambda_1 < L_2$. Then we have $\|\mathbf{z}_t - \mathbf{z}^*\| \leq D$ for all $t \geq 1$, where $D^2 = \frac{L_1^2}{\lambda_1^2} + \frac{2L_2^2}{\lambda_1^2}\|\mathbf{z}_1 - \mathbf{z}^*\|^2$. Moreover, it holds that*

$$\text{Gap}_{\mathcal{X}\times\mathcal{Y}}(\bar{\mathbf{z}}_{T+1}) \leq \frac{L_2\left(\sup_{\mathbf{z}\in\mathcal{X}\times\mathcal{Y}}\|\mathbf{z} - \mathbf{z}^*\|^2 + \frac{5}{4}D^2\right)\sqrt{\frac{8\|\mathbf{F}(\mathbf{z}_1)\|}{\lambda_1} + 145\|\mathbf{z}_1 - \mathbf{z}^*\|^2}}{T^{1.5}}, \quad (20)$$

$$\min_{t\in\{2,\ldots,T+1\}}\|\mathbf{F}(\mathbf{z}_t)\| \leq \frac{3L_2D\sqrt{\frac{4\|\mathbf{F}(\mathbf{z}_1)\|}{\lambda_1} + 72.5\|\mathbf{z}_1 - \mathbf{z}^*\|^2}}{T}. \quad (21)$$

Under the additional assumption of a Lipschitz operator, Theorem 6.2 guarantees that the iterates stay bounded. This is the main technical difficulty in the analysis, as most previous works on adaptive methods assume a compact set. On the contrary, we prove that iterates remain bounded within a set of diameter $D = \mathcal{O}(\frac{L_1}{\lambda_1} + \frac{L_2}{\lambda}\|\mathbf{z}_1 - \mathbf{z}^*\|)$. Compared to Option **(I)** in Theorem 6.1, the diameter increases by a factor of $\frac{L_2}{\lambda_1}$, i.e., the ratio between $L_2$ and our initial parameter $\lambda_1$. Moreover, Theorem 6.2 guarantees the same convergence rate of $\mathcal{O}(T^{-1.5})$. In terms of constants, compared to Theorem 6.1, the difference is no more than $(\frac{L_2}{\lambda_1})^{2.5}$. Thus, with a reasonable underestimate of the Lipschitz constant, $\lambda_1 = cL_2$ for some absolute constant $c < 1$, the bound worsens only by a constant factor.

*Proof Sketch of Theorem 6.2.* We begin with the convergence with respect to (Gap) in (20). The proof consists of the following three steps.

**Step 1:** By using Proposition 5.3, we first establish the following recursive inequality:

$$\|\mathbf{z}_{t+1} - \mathbf{z}^*\|^2 \leq \frac{L_1^2}{\lambda_t^2} + 2\|\mathbf{z}_1 - \mathbf{z}^*\|^2 + \sum_{s=2}^{t} \left(\frac{\lambda_s}{\lambda_{s-1}} - 1\right)^2 \|\mathbf{z}_s - \mathbf{z}^*\|^2 - \frac{1}{2}\sum_{s=1}^{t}\|\mathbf{z}_{s+1} - \mathbf{z}_s\|^2, \quad (22)$$

as shown in Lemma C.1 in the Appendix. Note that this upper bound for $\|\mathbf{z}_{t+1} - \mathbf{z}^*\|^2$ on the right-hand side depends on $\|\mathbf{z}_s - \mathbf{z}^*\|^2$ for all $s \leq t$. By analyzing this recursive relation, we obtain $\|\mathbf{z}_{t+1} - \mathbf{z}^*\| \leq D$ and $\sum_{s=0}^{t}\|\mathbf{z}_s - \mathbf{z}_{s+1}\|^2 \leq 2D^2$ for all $t \geq 1$, where $D^2 = \frac{L_1^2}{\lambda_1^2} + \frac{2L_2^2}{\lambda_1^2}\|\mathbf{z}_1 - \mathbf{z}^*\|^2$; see Lemma C.2 for details.

**Step 2:** After showing a uniform upper bound on $\|\mathbf{z}_{t+1} - \mathbf{z}^*\|$, Proposition C.3 establishes the adaptive convergence bound $\text{Gap}_{\mathcal{X}\times\mathcal{Y}}(\bar{\mathbf{z}}_{T+1}) \leq L_2\left(\sup_{\mathbf{z}\in\mathcal{X}\times\mathcal{Y}}\|\mathbf{z} - \mathbf{z}^*\|^2 + \frac{5}{4}D^2\right)\left(\sum_{t=0}^{T}\eta_t\right)^{-1}$.

**Step 3:** Following similar arguments as in the proof of Theorem 6.1, we can show $\sum_{t=1}^{T}\frac{1}{\eta_t^2} = \mathcal{O}(D^2 + \frac{1}{\lambda_1}\|\mathbf{F}(\mathbf{z}_1)\|)$ (check Lemma C.5). By applying the Hölder's inequality $\sum_{t=0}^{T}\eta_t \geq T^{1.5}(\sum_{t=0}^{T}(1/\eta_t^2))^{-1/2}$, we obtain the final convergence rate.

Finally, along the same lines as Theorem 6.1, we can show that $\sum_{t=2}^{T+1}\frac{1}{\lambda_t}\|\mathbf{F}(\mathbf{z}_t)\| = \mathcal{O}\left(D\sqrt{\sum_{t=1}^{T}\frac{1}{\eta_t^2}}\right) = \mathcal{O}\left(D\sqrt{D^2 + \frac{1}{\lambda_1}\|\mathbf{F}(\mathbf{z}_1)\|}\right)$. Since $\lambda_t \leq L_2$ and $\min_{\{2,\ldots,T+1\}}\|\mathbf{F}(\mathbf{z}_t)\| \leq \frac{1}{T}\sum_{t=2}^{T+1}\|\mathbf{F}(\mathbf{z}_t)\|$, we obtain the result in (21). $\qquad\square$

*Remark* 6.1. Our results can be extended to the more general problem of monotone inclusion with proper modification to the algorithm. We chose to focus on the unconstrained min-max problem for ease of presentation, so that we can better highlight the key novelties and make it accessible to a broader audience. In future work, we plan to extend our results for the monotone inclusion problem.

*Remark* 6.2. Our proposed algorithm with Option (**II**) achieves the same rate as Option (**I**) but does not require prior knowledge of the Hessian's Lipschitz constant, making it fully parameter-free. However, since it requires an additional assumption on Lipschitz gradients (Assumption 2.3), the existing lower bound [23] does not directly apply to certify its optimality. That said, we hypothesize that the $L_1$-Lipschitz gradient assumption should not improve the lower bound based on the existing evidence from the convex minimization setting. Specifically, [42] proves that for convex minimization with $L_1$-Lipschitz gradient and $L_2$-Lipschitz Hessian, the optimal rate is $\mathcal{O}\left(\min\left\{\frac{L_1 D^2}{T^2}, \frac{L_2 D^3}{T^{3.5}}\right\}\right)$, where $D$ is the initial distance. When $T$ is sufficiently large, the second term will become the smaller one, showing that the Lipschitz gradient assumption does not improve the optimal rate. While their construction does not imply an analogous rate for our setting in Theorem 6.2, we conjecture that the Lipschitz gradient assumption should not improve the lower bound of $\Omega(1/T^{1.5})$.

## 7 Numerical experiments

In this section, we present numerical results for implementing both variants of our algorithm: the version with $\lambda_t = L_2$ (Adaptive SOM I) and the parameter-free variant (Adaptive SOM II). We also compare these with the homotopy inexact proximal-Newton extragradient (HIPNEX) method [30] and the optimistic second-order method with line search (Optimal SOM) [21]. To assess convergence toward the solution $\mathbf{z}^*$, we plot $\|\mathbf{F}(\mathbf{z}_T)\|^2/\|\mathbf{F}(\mathbf{z}_0)\|^2$. For complete details, check Appendix D.

**Synthetic min-max problem:** We first consider the min-max problem in [21, 30], given by

$$\min_{\mathbf{x}\in\mathbb{R}^n}\max_{\mathbf{y}\in\mathbb{R}^n} f(\mathbf{x}, \mathbf{y}) = (\mathbf{A}\mathbf{x} - \mathbf{b})^\top\mathbf{y} + (L_2/6)\|\mathbf{x}\|^3,$$

which satisfies Assumptions 2.1 and 2.2. Let $\mathbf{z} = (\mathbf{x}, \mathbf{y}) \in \mathbb{R}^d$, with $d = 2n$, and recall that $\mathbf{F}(\mathbf{z})$ is defined in (2). Following the setup in [30], we generate the matrix $\mathbf{A} \in \mathbb{R}^{d\times d}$ to ensure a condition number of 20. The vector $\mathbf{b} \in \mathbb{R}^d$ is generated randomly according to $\mathcal{N}(0, \mathbf{I})$. We report results across various values of $L_2$ and problem dimension $d$ (for complete details, see Appendix D) and present a representative subset here. Focusing on large dimensions highlights computational efficiency, as shown in Fig. 1, where our line-search-free methods outperform both the optimal SOM and HIPNEX in runtime. The performance gap with the optimal SOM widens as the dimension grows: line search demands more steps, especially with larger $L_2$, with each step becoming increasingly costly due to the Hessian computation and inversion in high dimensions.

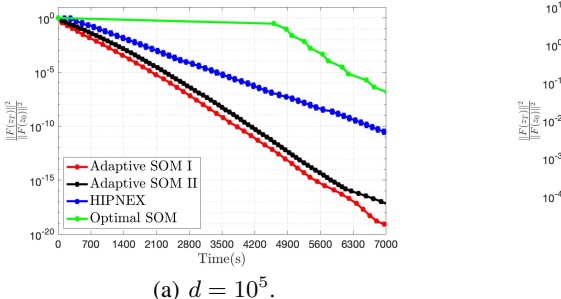 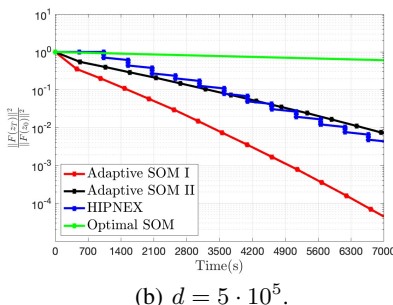

(a) $d = 10^5$.  (b) $d = 5 \cdot 10^5$.

Figure 1: Synthetic min-max problem: Runtimes under large dimension regime with $L_2 = 10^4$.

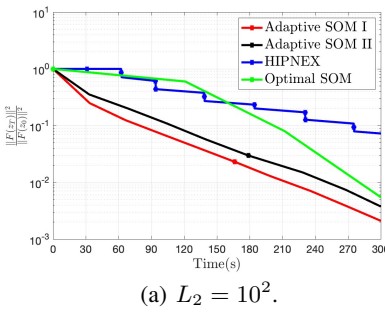 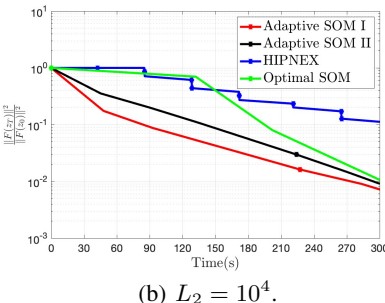

(a) $L_2 = 10^2$.  (b) $L_2 = 10^4$.

Figure 2: AUC maximization: Runtimes under large Lipschitz ($L_2$) regime with dimension $d = 10^4$.

**AUC maximization problem:** We consider a second problem where we maximize the Area Under the Receiver Operating Characteristic Curve (AUC), where we want to find a classifier $\boldsymbol{\theta} \in \mathbb{R}^d$ with a small error and a large AUC. This problem could be formulated as a min-max problem as in [43–45]:

$$\min_{\mathbf{x}=(\boldsymbol{\theta},u,v)} \max_{y} \frac{1-p}{N}\Big(\sum_{i=1}^{N}(\langle\boldsymbol{\theta},\mathbf{a}_i\rangle - u)^2\mathbb{I}\,[b_i = 1]\Big) + \frac{p}{N}\Big(\sum_{i=1}^{N}(\langle\boldsymbol{\theta},\mathbf{a}_i\rangle - v)^2\mathbb{I}\,[b_i = -1]\Big)$$

$$+ \frac{2(1+y)}{N}\Big(\sum_{i=1}^{N}\langle\boldsymbol{\theta},\mathbf{a}_i\rangle(p\mathbb{I}\,[b_i = -1] - (1-p)\mathbb{I}\,[b_i = 1])\Big) + \frac{\rho}{6}\|\mathbf{x}\|^3 - p(1-p)y^2,$$

where $u, v \in \mathbb{R}$ are auxiliary variables, $\{(\mathbf{a}_i, b_i)\}_{i=1}^{N}$ denote the (data, label) pairs ($\mathbf{a}_i \in \mathbb{R}^d$ and $b_i \in \{1, -1\}$), $\mathbb{I}\,[\cdot]$ is the indicator function, and $p$ is the ratio of positive labels. Similar to the observations above, Fig. 2 demonstrates that both of our methods outperform the optimal SOM and HIPNEX in terms of runtime, particularly in the early stages of the execution.

## 8 Conclusion and limitations

We proposed the first parameter-free and line-search-free second-order method for solving convex-concave min-max optimization problems. Our methods eliminate the need for line-search and backtracking mechanisms by identifying a sufficient condition on the approximation error and designing a data-adaptive update rule for step size $\eta_t$ that satisfies this condition. Notably, distinct from conventional approaches, our adaptive step size rule can be non-monotonic. Additionally, we removed the requirement to know the Lipschitz constant of the Hessian by appropriately regularizing the Hessian matrix with an adaptive scaling parameter $\lambda_t$.

The convergence rate for our fully parameter-free method was established under the additional assumption that the gradient is Lipschitz continuous. This assumption helps control the prediction error without imposing artificial boundedness conditions. Our method ensures that the generated sequence remains bounded even without access to any Lipschitz parameters. Extending these parameter-free guarantees without the Lipschitz gradient assumption remains an open problem worth exploring.

## Acknowledgments

This work was supported in part by the NSF CAREER Award CCF-2338846, the NSF AI Institute for Foundations of Machine Learning (IFML) and NSF Tripods ENCORE Institute. Research of Ali Kavis is funded in part by the Swiss National Science Foundation (SNSF) under grant number P500PT_217942.

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

# Appendix

## A    Missing proofs in Section 5

### A.1    Proofs of Propositions 5.1 and 5.3

Before proving Propositions 5.1 and 5.3, we first present a key lemma.

**Lemma A.1.** *Consider the update rule in* (6). *For any* $\mathbf{z} \in \mathbb{R}^d$, *we have*

$$
\eta_t \langle \mathbf{F}(\mathbf{z}_{t+1}), \mathbf{z}_{t+1} - \mathbf{z} \rangle = \eta_t \langle \mathbf{e}_{t+1}, \mathbf{z}_{t+1} - \mathbf{z} \rangle - \eta_{t-1} \langle \mathbf{e}_t, \mathbf{z}_t - \mathbf{z} \rangle + \eta_{t-1} \langle \mathbf{e}_t, \mathbf{z}_t - \mathbf{z}_{t+1} \rangle
$$
$$
+ \frac{\lambda_t}{2} \left( \|\mathbf{z}_t - \mathbf{z}\|^2 - \|\mathbf{z}_{t+1} - \mathbf{z}\|^2 - \|\mathbf{z}_t - \mathbf{z}_{t+1}\|^2 \right). \tag{23}
$$

*Proof.* To begin with, we rewrite the update rule in (6) in the following equivalent form:

$$
\mathbf{z}_{t+1} = \mathbf{z}_t - (\lambda_t \mathbf{I} + \eta_t \mathbf{F}'(\mathbf{z}_t))^{-1} (\eta_t \mathbf{F}(\mathbf{z}_t) + \eta_{t-1} \mathbf{e}_t)
$$
$$
\Leftrightarrow \quad (\lambda_t \mathbf{I} + \eta_t \mathbf{F}'(\mathbf{z}_t))(\mathbf{z}_{t+1} - \mathbf{z}_t) = -\eta_t \mathbf{F}(\mathbf{z}_t) - \eta_{t-1} \mathbf{e}_t
$$
$$
\Leftrightarrow \quad \eta_t (\mathbf{F}(\mathbf{z}_t) + \mathbf{F}'(\mathbf{z}_t)(\mathbf{z}_{t+1} - \mathbf{z}_t)) = \lambda_t (\mathbf{z}_t - \mathbf{z}_{t+1}) - \eta_{t-1} \mathbf{e}_t.
$$

Hence, by using the definition $\mathbf{e}_{t+1} = \mathbf{F}(\mathbf{z}_{t+1}) - \mathbf{F}(\mathbf{z}_t) - \mathbf{F}'(\mathbf{z}_t)(\mathbf{z}_{t+1} - \mathbf{z}_t)$, this further implies that

$$
\eta_t \mathbf{F}(\mathbf{z}_{t+1}) = \eta_t \mathbf{e}_{t+1} + \eta_t (\mathbf{F}(\mathbf{z}_t) + \mathbf{F}'(\mathbf{z}_t)(\mathbf{z}_{t+1} - \mathbf{z}_t)) = \eta_t \mathbf{e}_{t+1} - \eta_{t-1} \mathbf{e}_t + \lambda_t (\mathbf{z}_t - \mathbf{z}_{t+1}). \tag{24}
$$

Moreover, we have

$$
\eta_t \langle \mathbf{F}(\mathbf{z}_{t+1}), \mathbf{z}_{t+1} - \mathbf{z} \rangle = \eta_t \langle \mathbf{e}_{t+1}, \mathbf{z}_{t+1} - \mathbf{z} \rangle - \eta_{t-1} \langle \mathbf{e}_t, \mathbf{z}_{t+1} - \mathbf{z} \rangle + \lambda_t \langle \mathbf{z}_t - \mathbf{z}_{t+1}, \mathbf{z}_{t+1} - \mathbf{z} \rangle
$$
$$
= \eta_t \langle \mathbf{e}_{t+1}, \mathbf{z}_{t+1} - \mathbf{z} \rangle - \eta_{t-1} \langle \mathbf{e}_t, \mathbf{z}_t - \mathbf{z} \rangle + \eta_{t-1} \langle \mathbf{e}_t, \mathbf{z}_t - \mathbf{z}_{t+1} \rangle
$$
$$
+ \frac{\lambda_t}{2} \left( \|\mathbf{z}_t - \mathbf{z}\|^2 - \|\mathbf{z}_{t+1} - \mathbf{z}\|^2 - \|\mathbf{z}_t - \mathbf{z}_{t+1}\|^2 \right),
$$

where we used the elementary equality $\langle \mathbf{a}, \mathbf{b} \rangle = \frac{1}{2}\|\mathbf{a} + \mathbf{b}\|^2 - \frac{1}{2}\|\mathbf{a}\|^2 - \frac{1}{2}\|\mathbf{b}\|^2$ in the last equality. This completes the proof.

$\square$

**Proof of Proposition 5.1.** By summing the inequality in (23) from $t = 1$ to $t = T$ and noting that the first two terms on the right-hand side telescope, we obtain:

$$
\sum_{t=1}^{T} \eta_t \langle \mathbf{F}(\mathbf{z}_{t+1}), \mathbf{z}_{t+1} - \mathbf{z} \rangle = \eta_T \langle \mathbf{e}_{T+1}, \mathbf{z}_{T+1} - \mathbf{z} \rangle - \eta_0 \langle \mathbf{e}_1, \mathbf{z}_1 - \mathbf{z} \rangle + \sum_{t=1}^{T} \eta_{t-1} \langle \mathbf{e}_t, \mathbf{z}_t - \mathbf{z}_{t+1} \rangle
$$
$$
+ \sum_{t=1}^{T} \left( \frac{\lambda_t}{2} \left( \|\mathbf{z}_t - \mathbf{z}\|^2 - \|\mathbf{z}_{t+1} - \mathbf{z}\|^2 - \|\mathbf{z}_t - \mathbf{z}_{t+1}\|^2 \right) \right).
$$

Since $\eta_0 = 0$, rearranging the terms lead to (9).

**Proof of Proposition 5.3.** we first note that, since $\mathbf{F}(\mathbf{z}^*) = 0$ and $\mathbf{F}$ is monotone, it holds that

$$
\langle \mathbf{F}(\mathbf{z}_{t+1}), \mathbf{z}_{t+1} - \mathbf{z}^* \rangle = \langle \mathbf{F}(\mathbf{z}_{t+1}) - \mathbf{F}(\mathbf{z}^*), \mathbf{z}_{t+1} - \mathbf{z}^* \rangle \geq 0.
$$

Moreover, dividing both sides of (23) by $\lambda_t$ and letting $\mathbf{z} = \mathbf{z}^*$, we obtain that

$$
0 \leq \frac{\eta_t}{\lambda_t} \langle \mathbf{F}(\mathbf{z}_{t+1}), \mathbf{z}_{t+1} - \mathbf{z}^* \rangle = \frac{\eta_t}{\lambda_t} \langle \mathbf{e}_{t+1}, \mathbf{z}_{t+1} - \mathbf{z}^* \rangle - \frac{\eta_{t-1}}{\lambda_t} \langle \mathbf{e}_t, \mathbf{z}_t - \mathbf{z}^* \rangle + \frac{\eta_{t-1}}{\lambda_t} \langle \mathbf{e}_t, \mathbf{z}_t - \mathbf{z}_{t+1} \rangle
$$
$$
+ \frac{1}{2} \left( \|\mathbf{z}_t - \mathbf{z}^*\|^2 - \|\mathbf{z}_{t+1} - \mathbf{z}^*\|^2 - \|\mathbf{z}_t - \mathbf{z}_{t+1}\|^2 \right).
$$

Rearranging the terms, we get

$$
\frac{1}{2} \|\mathbf{z}_{t+1} - \mathbf{z}^*\|^2 \leq \frac{1}{2} \|\mathbf{z}_t - \mathbf{z}^*\|^2 + \frac{\eta_t}{\lambda_t} \langle \mathbf{e}_{t+1}, \mathbf{z}_{t+1} - \mathbf{z}^* \rangle - \frac{\eta_{t-1}}{\lambda_t} \langle \mathbf{e}_t, \mathbf{z}_t - \mathbf{z}^* \rangle
$$
$$
+ \frac{\eta_{t-1}}{\lambda_t} \langle \mathbf{e}_t, \mathbf{z}_t - \mathbf{z}_{t+1} \rangle - \frac{1}{2} \|\mathbf{z}_t - \mathbf{z}_{t+1}\|^2.
$$

By summing the above inequality from $t = 1$ to $t = T$, we obtain that

$$
\begin{aligned}
\frac{1}{2}\|\mathbf{z}_{T+1} - \mathbf{z}^*\|^2 \leq \frac{1}{2}\|\mathbf{z}_1 - \mathbf{z}^*\|^2 &- \sum_{t=1}^{T} \frac{1}{2}\|\mathbf{z}_t - \mathbf{z}_{t+1}\|^2 + \sum_{t=1}^{T} \frac{\eta_{t-1}}{\lambda_t}\langle \mathbf{e}_t, \mathbf{z}_t - \mathbf{z}_{t+1}\rangle \\
&+ \sum_{t=1}^{T} \left( \frac{\eta_t}{\lambda_t}\langle \mathbf{e}_{t+1}, \mathbf{z}_{t+1} - \mathbf{z}^*\rangle - \frac{\eta_{t-1}}{\lambda_t}\langle \mathbf{e}_t, \mathbf{z}_t - \mathbf{z}^*\rangle \right)
\end{aligned}
\tag{25}
$$

Finally, we can write

$$
\frac{\eta_t}{\lambda_t}\langle \mathbf{e}_{t+1}, \mathbf{z}_{t+1} - \mathbf{z}^*\rangle - \frac{\eta_{t-1}}{\lambda_t}\langle \mathbf{e}_t, \mathbf{z}_t - \mathbf{z}^*\rangle
$$

$$
= \frac{\eta_t}{\lambda_t}\langle \mathbf{e}_{t+1}, \mathbf{z}_{t+1} - \mathbf{z}^*\rangle - \frac{\eta_{t-1}}{\lambda_{t-1}}\langle \mathbf{e}_t, \mathbf{z}_t - \mathbf{z}^*\rangle + \left( \frac{1}{\lambda_{t-1}} - \frac{1}{\lambda_t} \right)\eta_{t-1}\langle \mathbf{e}_t, \mathbf{z}_t - \mathbf{z}^*\rangle.
$$

Summing the above inequality from $t = 1$ to $t = T$ yields

$$
\sum_{t=1}^{T} \left( \frac{\eta_t}{\lambda_t}\langle \mathbf{e}_{t+1}, \mathbf{z}_{t+1} - \mathbf{z}^*\rangle - \frac{\eta_{t-1}}{\lambda_t}\langle \mathbf{e}_t, \mathbf{z}_t - \mathbf{z}^*\rangle \right)
$$

$$
= \frac{\eta_T}{\lambda_T}\langle \mathbf{e}_{T+1}, \mathbf{z}_{T+1} - \mathbf{z}^*\rangle + \sum_{t=2}^{T} \left( \frac{1}{\lambda_{t-1}} - \frac{1}{\lambda_t} \right)\eta_{t-1}\langle \mathbf{e}_t, \mathbf{z}_t - \mathbf{z}^*\rangle
$$

The inequality in (13) follows by combining (25) and the above inequality.

### A.2 Proof of Lemma 5.2

We first rewrite the update rule in (6) in the following equivalent form:

$$
\mathbf{z}_{t+1} = \mathbf{z}_t - (\lambda_t\mathbf{I} + \eta_t\mathbf{F}'(\mathbf{z}_t))^{-1}(\eta_t\mathbf{F}(\mathbf{z}_t) + \eta_{t-1}\mathbf{e}_t)
$$

$$
\Leftrightarrow \quad (\lambda_t\mathbf{I} + \eta_t\mathbf{F}'(\mathbf{z}_t))(\mathbf{z}_{t+1} - \mathbf{z}_t) = -\eta_t\mathbf{F}(\mathbf{z}_t) - \eta_{t-1}\mathbf{e}_t.
$$

By taking the inner product with $\mathbf{z}_{t+1} - \mathbf{z}_t$ for both sides of the equaltiy, we obtain that

$$
\lambda_t\|\mathbf{z}_{t+1} - \mathbf{z}_t\|^2 + \eta_t\langle \mathbf{F}'(\mathbf{z}_t)(\mathbf{z}_{t+1} - \mathbf{z}_t), \mathbf{z}_{t+1} - \mathbf{z}_t\rangle = -\langle \eta_t\mathbf{F}(\mathbf{z}_t) + \eta_{t-1}\mathbf{e}_t, \mathbf{z}_{t+1} - \mathbf{z}_t\rangle. \tag{26}
$$

Since $\mathbf{F}$ is monotone by Assumption 2.1, this implies that the Jacobian matrix $\mathbf{F}'(\mathbf{z}_t)$ satisifes $\langle \mathbf{F}'(\mathbf{z}_t)\mathbf{z}, \mathbf{z}\rangle \geq 0$ for any $\mathbf{z} \in \mathbb{R}^m \times \mathbb{R}^n$ (e.g., see [46, Section 2].) Thus, we have $\langle \mathbf{F}'(\mathbf{z}_t)(\mathbf{z}_{t+1} - \mathbf{z}_t), \mathbf{z}_{t+1} - \mathbf{z}_t\rangle \geq 0$ and (26) further implies that

$$
\lambda_t\|\mathbf{z}_{t+1} - \mathbf{z}_t\|^2 \leq -\langle \eta_t\mathbf{F}(\mathbf{z}_t) + \eta_{t-1}\mathbf{e}_t, \mathbf{z}_{t+1} - \mathbf{z}_t\rangle \leq \|\eta_t\mathbf{F}(\mathbf{z}_t) + \eta_{t-1}\mathbf{e}_t\|\|\mathbf{z}_{t+1} - \mathbf{z}_t\|.
$$

Hence, we obtain that $\|\mathbf{z}_{t+1} - \mathbf{z}_t\| \leq \frac{1}{\lambda_t}\|\eta_t\mathbf{F}(\mathbf{z}_t) + \eta_{t-1}\mathbf{e}_t\| \leq \frac{1}{\lambda_t}\eta_t\|\mathbf{F}(\mathbf{z}_t)\| + \frac{1}{\lambda_t}\eta_{t-1}\|\mathbf{e}_t\|$ from the triangle inequality.

## B   Proof of Theorem 6.1

We first present the following key proposition, which will be the cornerstone of our convergence analysis. We establish that the iterates remain within a neighborhood of a solution characterized by the initial distance (part (a)) and that optimization path has finite length (part (c)). We also present the adaptive convergence bound (part (b)). The proof is in Appendix B.1.

**Proposition B.1.** *Suppose Assumptions 2.1 and 2.2 hold and let $\{\mathbf{z}_t\}_{t=0}^{T+1}$ be generated by Algorithm 1, where $\lambda_t = L_2$ (**Option I**) and $\alpha \in (0, \frac{1}{2})$. Then the following results hold:*

(a) $\|\mathbf{z}_t - \mathbf{z}^*\|^2 \leq \frac{1}{1-\alpha}\|\mathbf{z}_1 - \mathbf{z}^*\|^2$ *for all $t \geq 1$.*

(b) *Consider the weighted average iterate $\bar{\mathbf{z}}_{T+1} := \sum_{t=1}^{T} \eta_t\mathbf{z}_{t+1}/(\sum_{t=1}^{T} \eta_t)$. For any compact sets $\mathcal{X} \subset \mathbb{R}^m$, $\mathcal{Y} \subset \mathbb{R}^n$, we have $\mathrm{Gap}_{\mathcal{X} \times \mathcal{Y}}(\bar{\mathbf{z}}_{T+1}) \leq \frac{L_2}{2} \sup_{\mathbf{z} \in \mathcal{X} \times \mathcal{Y}} \|\mathbf{z}_1 - \mathbf{z}\|^2 \left( \sum_{t=1}^{T} \eta_t \right)^{-1}$*

*(c)* $\sum_{t=1}^{T} \|\mathbf{z}_t - \mathbf{z}_{t+1}\|^2 \leq \frac{1}{1-2\alpha} \|\mathbf{z}_1 - \mathbf{z}^*\|^2.$

In Proposition B.1, we have shown that $\sum_{t=0}^{T} \|\mathbf{z}_{t+1} - \mathbf{z}_t\|^2$ is bounded. Using that, our goal is to express the upper bound on $\frac{1}{\eta_t^2}$ in terms of $\|\mathbf{z}_{t+1} - \mathbf{z}_t\|$, which will help us show that $\frac{1}{\eta_t^2}$ is a summable sequence. This will verify that we achieve the optimal rate of $\mathcal{O}(1/T^{1.5})$. We begin by computing upper bounds on $\frac{1}{\eta_t^2}$ and $\|\mathbf{F}(\mathbf{z}_t)\|$ in the following lemma, whose proof can be found in Appendix B.2.

**Lemma B.2.** *For* $t \geq 1$*, the following results hold:*

*(a)* $\frac{1}{\eta_t^2} \leq \frac{1}{4}\|\mathbf{z}_t - \mathbf{z}_{t-1}\|^2 + \frac{1}{\alpha\lambda}\|\mathbf{F}(\mathbf{z}_t)\|;$

*(b)* $\|\mathbf{F}(\mathbf{z}_{t+1})\| \leq \frac{(1+\alpha)\lambda}{\eta_t}\|\mathbf{z}_{t+1} - \mathbf{z}_t\| + \frac{\alpha\lambda}{\eta_t}\|\mathbf{z}_t - \mathbf{z}_{t-1}\|.$

Using the bounds established in Lemma B.2, we prove an upper bound on $\sum_{t=1}^{T} \frac{1}{\eta_t^2}$ as in the following lemma. The proof is in Appendix B.3.

**Lemma B.3.** *We have*

$$\sum_{t=1}^{T} \frac{1}{\eta_t^2} \leq \frac{17\alpha^2 + 16\alpha + 4}{2(1-2\alpha)\alpha^2} \|\mathbf{z}_1 - \mathbf{z}^*\|^2 + \frac{2}{\alpha\lambda}\|\mathbf{F}(\mathbf{z}_1)\|.$$

Now we are ready to prove Theorem 6.1. Besides the convergence bound in terms of the (Gap) function, we provide an additional bound with respect to the norm of the operator, evaluated at the "best" iterate.

*Proof of Theorem 6.1.* By Proposition B.1,

$$\mathrm{Gap}(\bar{\mathbf{z}}_{T+1}) \leq \left( \sum_{t=0}^{T} \eta_t \right)^{-1} \frac{1}{2\lambda} \sup_{\mathbf{z} \in \mathcal{X} \times \mathcal{Y}} \|\mathbf{z}_0 - \mathbf{z}^*\|^2.$$

Moreover by Holder's inequality we can show,

$$\sum_{t=0}^{T} \eta_t \geq T^{1.5} \left( \sum_{t=0}^{T} \frac{1}{\eta_t^2} \right)^{-1/2} \tag{27}$$

Plugging in the lower bound on $\sum_{t=0}^{T} \eta_t$ from (27) yields

$$\mathrm{Gap}(\bar{\mathbf{z}}_{T+1}) \leq \frac{\frac{1}{2} \sup_{\mathbf{z} \in \mathcal{X} \times \mathcal{Y}} \|\mathbf{z}_0 - \mathbf{z}^*\|^2 \cdot \sqrt{\sum_{t=0}^{T} \frac{1}{\eta_t^2}}}{T^{1.5}}.$$

Combining the above with the upper bound in Lemma B.3 completes the result.

Next we prove the complementary convergence bound with respect to the norm of the operator. From (36) (see the proof of Lemma B.3), we also obtain that

$$\sum_{t=2}^{T+1} \|\mathbf{F}(\mathbf{z}_t)\| \leq \frac{(1+2\alpha)\lambda\|\mathbf{z}_1 - \mathbf{z}^*\|}{\sqrt{1-2\alpha}} \sqrt{\sum_{t=1}^{T} \frac{1}{\eta_t^2}}. \tag{28}$$

Combining (28) with Lemma B.3, we obtain that

$$\sum_{t=2}^{T+1} \|\mathbf{F}(\mathbf{z}_t)\| \leq \frac{(1+2\alpha)\lambda\|\mathbf{z}_1 - \mathbf{z}^*\|}{\sqrt{1-2\alpha}} \sqrt{\frac{\|\mathbf{z}_1 - \mathbf{z}^*\|^2}{2(1-2\alpha)} + \frac{2(1+2\alpha)^2\|\mathbf{z}_1 - \mathbf{z}^*\|^2}{\alpha^2(1-2\alpha)} + \frac{2}{\alpha\lambda}\|\mathbf{F}(\mathbf{z}_1)\|}.$$

Finally, the result follows from the fact that $\min_{t \in \{2,\ldots,T+1\}} \|\mathbf{F}(\mathbf{z}_t)\| \leq \frac{1}{T} \sum_{t=2}^{T+1} \|\mathbf{F}(\mathbf{z}_t)\|.$

$\square$

## B.1 Proof of Proposition B.1

Before proving Proposition B.1, we will formalize the error condition implied by the constant choice of regularization parameter $\lambda_t = \lambda$.

**Lemma B.4.** *Consider the update rule in (6) and let $\eta_t$ be given by (7). Then we have $\eta_t \|\mathbf{z}_{t+1} - \mathbf{z}_t\| \leq 2\alpha$ and $\eta_t \|\mathbf{e}_{t+1}\| \leq \alpha\lambda \|\mathbf{z}_{t+1} - \mathbf{z}_t\|$.*

*Proof.* We first prove that $\eta_t \|\mathbf{z}_{t+1} - \mathbf{z}_t\| \leq 2\alpha$. To see this, we define $\eta_t$ as (7) by solving the following quadratic equation:

$$\eta_t(\eta_t \|\mathbf{F}(\mathbf{z}_t)\| + \eta_{t-1}\|\mathbf{e}_t\|) = \frac{2\alpha\lambda^2}{L_2}.$$

Thus, by using Lemma 5.2 with $\lambda_t = \lambda = L_2$, we can prove that

$$\eta_t \|\mathbf{z}_{t+1} - \mathbf{z}_t\| \leq \frac{\eta_t}{\lambda}(\eta_t \|\mathbf{F}(\mathbf{z}_t)\| + \eta_{t-1}\|\mathbf{e}_t\|) \leq 2\alpha.$$

Note that $\|\mathbf{e}_{t+1}\| := \|\mathbf{F}(\mathbf{z}_{t+1}) - \mathbf{F}(\mathbf{z}_t) - \mathbf{F}'(\mathbf{z}_t)(\mathbf{z}_{t+1} - \mathbf{z}_t)\| \leq \frac{L_2}{2}\|\mathbf{z}_{t+1} - \mathbf{z}_t\|^2 = \frac{\lambda}{2}\|\mathbf{z}_{t+1} - \mathbf{z}_t\|^2$ by Assumption 2.2. Hence, this implies that $\eta_t \|\mathbf{e}_{t+1}\| \leq \frac{\lambda\eta_t}{2}\|\mathbf{z}_{t+1} - \mathbf{z}_t\|^2 \leq \alpha\lambda \|\mathbf{z}_{t+1} - \mathbf{z}_t\|$. $\qquad\square$

Now, we have all the necessary tools to prove Proposition B.1.

***Proof of Proposition B.1.*** We first use Lemma B.4 to control the error terms in (9) and (13). Specifically, by using Cauchy-Schwarz inequality and Young's inequality, for $t \geq 2$ we obtain:

$$\eta_{t-1}\langle \mathbf{e}_t, \mathbf{z}_t - \mathbf{z}_{t+1}\rangle \leq \eta_{t-1}\|\mathbf{e}_t\|\|\mathbf{z}_t - \mathbf{z}_{t+1}\| \leq \alpha\lambda\|\mathbf{z}_t - \mathbf{z}_{t-1}\|\|\mathbf{z}_t - \mathbf{z}_{t+1}\|$$
$$\leq \frac{\alpha\lambda}{2}\|\mathbf{z}_t - \mathbf{z}_{t-1}\|^2 + \frac{\alpha\lambda}{2}\|\mathbf{z}_t - \mathbf{z}_{t+1}\|^2. \tag{29}$$

Similarly, we can bound the first term by

$$\eta_T\langle \mathbf{e}_{T+1}, \mathbf{z}_{T+1} - \mathbf{z}\rangle \leq \eta_T\|\mathbf{e}_{T+1}\|\|\mathbf{z}_{T+1} - \mathbf{z}\| \leq \frac{\alpha\lambda}{2}\|\mathbf{z}_{T+1} - \mathbf{z}_T\|^2 + \frac{\alpha\lambda}{2}\|\mathbf{z}_{T+1} - \mathbf{z}\|^2. \tag{30}$$

**Proof of (a)** Since $\lambda_t = \lambda$ for all $t \geq 1$, the first summation term on the right-hand side of (9) telescope:

$$\sum_{t=1}^{T} \frac{\lambda}{2}\left(\|\mathbf{z}_t - \mathbf{z}\|^2 - \|\mathbf{z}_{t+1} - \mathbf{z}\|^2\right) = \frac{\lambda}{2}\|\mathbf{z}_1 - \mathbf{z}\|^2 - \frac{\lambda}{2}\|\mathbf{z}_{T+1} - \mathbf{z}\|^2.$$

Furthermore, by (29) and (30), we have

$$\eta_T\langle \mathbf{e}_{T+1}, \mathbf{z}_{T+1} - \mathbf{z}\rangle + \sum_{t=2}^{T} \eta_{t-1}\langle \mathbf{e}_t, \mathbf{z}_t - \mathbf{z}_{t+1}\rangle$$

$$\leq \frac{\alpha\lambda}{2}\|\mathbf{z}_{T+1} - \mathbf{z}_T\|^2 + \frac{\alpha\lambda}{2}\|\mathbf{z}_{T+1} - \mathbf{z}\|^2 + \sum_{t=2}^{T}\left(\frac{\alpha\lambda}{2}\|\mathbf{z}_t - \mathbf{z}_{t-1}\|^2 + \frac{\alpha\lambda}{2}\|\mathbf{z}_t - \mathbf{z}_{t+1}\|^2\right) \tag{31}$$

$$\leq \frac{\alpha\lambda}{2}\|\mathbf{z}_{T+1} - \mathbf{z}\|^2 + \alpha\lambda\sum_{t=1}^{T}\|\mathbf{z}_t - \mathbf{z}_{t+1}\|^2.$$

Hence, by applying all the inequalities above in (9), we obtain that

$$\sum_{t=1}^{T}\eta_t\langle \mathbf{F}(\mathbf{z}_{t+1}), \mathbf{z}_{t+1} - \mathbf{z}\rangle \leq \frac{\lambda}{2}\|\mathbf{z}_1 - \mathbf{z}\|^2 - \frac{(1-\alpha)\lambda}{2}\|\mathbf{z}_{T+1} - \mathbf{z}\|^2 - \sum_{t=1}^{T}\frac{(1-2\alpha)\lambda}{2}\|\mathbf{z}_t - \mathbf{z}_{t+1}\|^2.$$

Since we have $\alpha \in (0, \frac{1}{2})$, the last two terms in the above inequality are negative and this further implies that $\sum_{t=1}^{T}\eta_t\langle \mathbf{F}(\mathbf{z}_{t+1}), \mathbf{z}_{t+1} - \mathbf{z}\rangle \leq \frac{\lambda}{2}\|\mathbf{z}_1 - \mathbf{z}\|^2 = \frac{L_1}{2}\|\mathbf{z}_1 - \mathbf{z}\|^2$. By applying Lemma 2.1, it leads to

$$f(\bar{\mathbf{x}}_{T+1}, \mathbf{y}) - f(\mathbf{x}, \bar{\mathbf{y}}_{T+1}) \leq \frac{\sum_{t=1}^{T}\eta_t\langle \mathbf{F}(\mathbf{z}_{t+1}), \mathbf{z}_{t+1} - \mathbf{z}\rangle}{\sum_{t=1}^{T}\eta_t} \leq \frac{L_1}{2}\|\mathbf{z}_1 - \mathbf{z}\|^2\left(\sum_{t=1}^{T}\eta_t\right)^{-1}.$$

Taking the supremum of $\mathbf{z} = (\mathbf{x}, \mathbf{y})$ over $\mathcal{X} \times \mathcal{Y}$, we obtain the desired result.

**Proof of (b) and (c)**    Since $\lambda_t = \lambda$ for all $t \geq 1$, the first summation term on the right-hand side of (13) telescope:

$$\sum_{t=1}^{T} \left( \frac{\eta_t}{\lambda} \langle \mathbf{e}_{t+1}, \mathbf{z}_{t+1} - \mathbf{z}^* \rangle - \frac{\eta_{t-1}}{\lambda} \langle \mathbf{e}_t, \mathbf{z}_t - \mathbf{z}^* \rangle \right) = \frac{\eta_T}{\lambda} \langle \mathbf{e}_{T+1}, \mathbf{z}_{T+1} - \mathbf{z}^* \rangle,$$

where we used the fact that $\eta_0 = 0$. Using (31), we also have

$$\frac{\eta_T}{\lambda} \langle \mathbf{e}_{T+1}, \mathbf{z}_{T+1} - \mathbf{z}^* \rangle + \sum_{t=2}^{T} \frac{\eta_{t-1}}{\lambda} \langle \mathbf{e}_t, \mathbf{z}_t - \mathbf{z}_{t+1} \rangle \leq \frac{\alpha}{2} \|\mathbf{z}_{T+1} - \mathbf{z}^*\|^2 + \alpha \sum_{t=1}^{T} \|\mathbf{z}_t - \mathbf{z}_{t+1}\|^2$$

Hence, applying the above inequality in (13), we obtain:

$$\frac{1-\alpha}{2} \|\mathbf{z}_{T+1} - \mathbf{z}^*\|^2 \leq \frac{1}{2} \|\mathbf{z}_1 - \mathbf{z}^*\|^2 - \frac{1-2\alpha}{2} \sum_{t=1}^{T} \|\mathbf{z}_t - \mathbf{z}_{t+1}\|^2. \tag{32}$$

To begin with, since $\alpha < \frac{1}{2}$, the last summation term in (32) is negative. Hence, this further implies that $\frac{1-\alpha}{2} \|\mathbf{z}_{T+1} - \mathbf{z}^*\|^2 \leq \frac{1}{2} \|\mathbf{z}_1 - \mathbf{z}^*\|^2$, which proves Part (b). Moreover, since the left-hand side of (32) is non-negative, this also leads to $\frac{1-2\alpha}{2} \sum_{t=1}^{T} \|\mathbf{z}_t - \mathbf{z}_{t+1}\|^2 \leq \frac{1}{2} \|\mathbf{z}_1 - \mathbf{z}^*\|^2$, which proves Part (c).

$\square$

## B.2   Proof of Lemma B.2

By the update rule in (8), we have

$$\begin{aligned}
\frac{1}{\eta_t^2} &= \frac{1}{16\alpha^2\lambda^2} \left( \eta_{t-1}\|\mathbf{e}_t\| + \sqrt{\eta_{t-1}^2\|\mathbf{e}_t\|^2 + 8\alpha\lambda\|\mathbf{F}(\mathbf{z}_t)\|} \right)^2 \\
&\leq \frac{1}{8\alpha^2\lambda^2} \left( \eta_{t-1}^2\|\mathbf{e}_t\|^2 + \eta_{t-1}^2\|\mathbf{e}_t\|^2 + 8\alpha\lambda\|\mathbf{F}(\mathbf{z}_t)\| \right) \\
&= \frac{\eta_{t-1}^2\|\mathbf{e}_t\|^2}{4\alpha^2\lambda^2} + \frac{\|\mathbf{F}(\mathbf{z}_t)\|}{\alpha\lambda} \leq \frac{1}{4}\|\mathbf{z}_t - \mathbf{z}_{t-1}\|^2 + \frac{\|\mathbf{F}(\mathbf{z}_t)\|}{\alpha\lambda}.
\end{aligned}$$

By using (6), we can write

$$\eta_t \mathbf{F}(\mathbf{z}_{t+1}) = \eta_t \mathbf{e}_{t+1} - \eta_{t-1}\mathbf{e}_t - \lambda(\mathbf{z}_{t+1} - \mathbf{z}_t).$$

Hence, by using the triangle inequality, we have

$$\eta_t\|\mathbf{F}(\mathbf{z}_{t+1})\| \leq \eta_t\|\mathbf{e}_{t+1}\| + \eta_{t-1}\|\mathbf{e}_t\| + \lambda\|\mathbf{z}_{t+1} - \mathbf{z}_t\| \leq (1+\alpha)\lambda\|\mathbf{z}_{t+1} - \mathbf{z}_t\| + \alpha\lambda\|\mathbf{z}_t - \mathbf{z}_{t-1}\|,$$

where we used Lemma B.4 in the last inequality.

## B.3   Proof of Lemma B.3

By summing the inequality in Part (a) in Lemma B.2 over $t = 1, \ldots, T$, we have

$$\sum_{t=1}^{T} \frac{1}{\eta_t^2} \leq \frac{1}{4} \sum_{t=1}^{T} \|\mathbf{z}_t - \mathbf{z}_{t-1}\|^2 + \frac{1}{\alpha\lambda} \sum_{t=1}^{T} \|\mathbf{F}(\mathbf{z}_t)\| \tag{33}$$

The first summation term can be bounded as $\frac{1}{4} \sum_{t=1}^{T} \|\mathbf{z}_t - \mathbf{z}_{t-1}\|^2 \leq \frac{1}{4(1-2\alpha)} \|\mathbf{z}_1 - \mathbf{z}^*\|^2$. For the second summation, we use Part (b) in Lemma B.2 to get

$$\sum_{t=1}^{T} \|\mathbf{F}(\mathbf{z}_t)\| \leq \|\mathbf{F}(\mathbf{z}_1)\| + \sum_{t=1}^{T-1} \left( \frac{(1+\alpha)\lambda}{\eta_t} \|\mathbf{z}_{t+1} - \mathbf{z}_t\| + \frac{\alpha\lambda}{\eta_t} \|\mathbf{z}_t - \mathbf{z}_{t-1}\| \right) \tag{34}$$

Further, it follows from Cauchy-Schwarz inequality that

$$\sum_{t=1}^{T-1} \frac{1}{\eta_t} \|\mathbf{z}_{t+1} - \mathbf{z}_t\| \leq \sqrt{\sum_{t=1}^{T-1} \frac{1}{\eta_t^2}} \sqrt{\sum_{t=1}^{T-1} \|\mathbf{z}_{t+1} - \mathbf{z}_t\|^2},$$

$$\sum_{t=1}^{T-1} \frac{1}{\eta_t} \|\mathbf{z}_t - \mathbf{z}_{t-1}\| \leq \sqrt{\sum_{t=1}^{T-1} \frac{1}{\eta_t^2}} \sqrt{\sum_{t=1}^{T-2} \|\mathbf{z}_{t+1} - \mathbf{z}_t\|^2}. \tag{35}$$

Since $\sum_{t=1}^{T} \|\mathbf{z}_t - \mathbf{z}_{t+1}\|^2 \leq \frac{1}{1-2\alpha} \|\mathbf{z}_1 - \mathbf{z}^*\|^2$ by Proposition B.1, combining (34) and (35) leads to

$$\sum_{t=1}^{T} \|\mathbf{F}(\mathbf{z}_t)\| \leq \|\mathbf{F}(\mathbf{z}_1)\| + \frac{(1+2\alpha)\lambda\|\mathbf{z}_1 - \mathbf{z}^*\|}{\sqrt{1-2\alpha}} \sqrt{\sum_{t=1}^{T-1} \frac{1}{\eta_t^2}}. \tag{36}$$

Plugging this bound back in (33), we arrive at

$$\sum_{t=1}^{T} \frac{1}{\eta_t^2} \leq \frac{1}{4(1-2\alpha)} \|\mathbf{z}_0 - \mathbf{z}^*\|^2 + \frac{1}{\alpha\lambda} \|\mathbf{F}(\mathbf{z}_1)\| + \frac{(1+2\alpha)\|\mathbf{z}_1 - \mathbf{z}^*\|}{\alpha\sqrt{1-2\alpha}} \sqrt{\sum_{t=1}^{T-1} \frac{1}{\eta_t^2}} \tag{37}$$

Note that $\sum_{t=0}^{T} \frac{1}{\eta_t^2}$ appears on both side of (37). To deal with this, we rely on the following lemma.

**Lemma B.5.** *Let $a, b \geq 0$ and suppose that $x \leq a + b\sqrt{x}$. Then it implies that $x \leq 2a + 2b^2$.*

*Proof.* We can rewrite the inequality as $(\sqrt{x} - \frac{b}{2})^2 \leq a + \frac{b^2}{4}$. Thus, $\sqrt{x} - \frac{b}{2} \leq \sqrt{a + \frac{b^2}{4}} \leq \sqrt{a} + \frac{b}{2}$, which leads to $\sqrt{x} \leq \sqrt{a} + b \implies x \leq (\sqrt{a} + b)^2 \leq 2a + 2b^2$. $\square$

Thus, by applying Lemma B.5, we obtain that

$$\sum_{t=1}^{T} \frac{1}{\eta_t^2} \leq \frac{1}{2(1-2\alpha)} \|\mathbf{z}_1 - \mathbf{z}^*\|^2 + \frac{2}{\alpha\lambda} \|\mathbf{F}(\mathbf{z}_1)\| + \frac{2(1+2\alpha)^2 \|\mathbf{z}_1 - \mathbf{z}^*\|^2}{\alpha^2(1-2\alpha)}$$

This completes the proof.

## C  Proof of Theorem 6.2

With the introduction of parameter-free $\eta_t$ and time-varying $\lambda_t$, one of the main requirements of the analysis is validating the boundedness of the iterate sequence $\{\mathbf{z}_t\}_{t=0}^{T+1}$ in the absence of the knowledge of $L_2$. Note that this is where we use the Lipschitz continuity of the gradient of $f$ (Assumption 2.3) to control the prediction error. We begin by an intermediate bound on the distance to a solution.

**Lemma C.1.** *Let $\alpha \in (0, \frac{1}{3})$. For any $t \geq 1$, it holds that*

$$\|\mathbf{z}_{t+1} - \mathbf{z}^*\|^2 \leq \frac{64\alpha^2 L_1^2}{\lambda_t^2} + 2\|\mathbf{z}_1 - \mathbf{z}^*\|^2 + \sum_{s=2}^{t} \left(\frac{\lambda_s}{\lambda_{s-1}} - 1\right)^2 \|\mathbf{z}_s - \mathbf{z}^*\|^2 \tag{38}$$

$$- 2(1 - 3\alpha) \sum_{s=1}^{t} \|\mathbf{z}_{s+1} - \mathbf{z}_s\|^2. \tag{39}$$

Based on the bound above, we present an analogue of the boundedness results in Proposition B.1 below.

**Lemma C.2.** *Define $D^2 = \frac{64\alpha^2 L_1^2}{\lambda_1^2} + \frac{2L_2^2}{\lambda_1^2} \|\mathbf{z}_1 - \mathbf{z}^*\|^2$. For any $t \geq 1$, we have*

$$\|\mathbf{z}_{t+1} - \mathbf{z}^*\| \leq D \quad and \quad \sum_{s=0}^{t} \|\mathbf{z}_s - \mathbf{z}_{s+1}\|^2 \leq \frac{1}{2(1-3\alpha)} D^2.$$

Now that we verified that the iterates remain bounded, we can state the adaptive convergence bound for the parameter-free algorithm.

**Proposition C.3.** *Suppose Assumptions 2.1 to 2.3 hold and let $\{\mathbf{z}_t\}_{t=0}^{T+1}$ be generated by Algorithm 1, where $\lambda_t$ is given by (8) (Option (II)) and $\alpha \in (0, \frac{1}{3})$. Define the averaged iterate $\bar{\mathbf{z}}_{T+1} = \sum_{t=0}^{T} \eta_t \mathbf{z}_{t+1} / (\sum_{t=0}^{T} \eta_t)$. Then we have*

$$\mathrm{Gap}_{\mathcal{X} \times \mathcal{Y}}(\bar{\mathbf{z}}_{T+1}) \leq L_2 \left( \sup_{\mathbf{z} \in \mathcal{X} \times \mathcal{Y}} \|\mathbf{z} - \mathbf{z}^*\|^2 + \left( \frac{9}{8} + \frac{\alpha^2}{4(1 - 3\alpha)} \right) D^2 \right) \left( \sum_{t=0}^{T} \eta_t \right)^{-1}.$$

In the sequel, we present the counterpart of Lemmas B.2 and B.3 for the parameter-free Option (II).

**Lemma C.4.** *For $t \geq 1$, the following results hold:*

(a) $\frac{1}{\eta_t^2} \leq \frac{1}{4} \|\mathbf{z}_t - \mathbf{z}_{t-1}\|^2 + \frac{1}{\alpha \lambda_t} \|\mathbf{F}(\mathbf{z}_t)\|$;

(b) $\|\mathbf{F}(\mathbf{z}_{t+1})\| \leq \frac{(1+\alpha)\lambda_t}{\eta_t} \|\mathbf{z}_{t+1} - \mathbf{z}_t\| + \frac{\alpha \lambda_t}{\eta_t} \|\mathbf{z}_t - \mathbf{z}_{t-1}\|$.

*Proof.* The proof follows from that of its analogue Lemma B.2 up to replacing $\lambda$ by $\lambda_t$. $\qquad\square$

**Lemma C.5.** *We have*

$$\sum_{t=0}^{T} \frac{1}{\eta_t^2} \leq \frac{17\alpha^2 + 16\alpha + 4}{4(1 - 3\alpha)\alpha^2} \|\mathbf{z}_1 - \mathbf{z}^*\|^2 + \frac{2}{\alpha \lambda_1} \|\mathbf{F}(\mathbf{z}_1)\|.$$

*Proof.* By summing the inequality in Part (a) in Lemma C.4 over $t = 1, \ldots, T$, we have

$$\sum_{t=1}^{T} \frac{1}{\eta_t^2} \leq \frac{1}{4} \sum_{t=1}^{T} \|\mathbf{z}_t - \mathbf{z}_{t-1}\|^2 + \frac{1}{\alpha} \sum_{t=1}^{T} \frac{1}{\lambda_t} \|\mathbf{F}(\mathbf{z}_t)\| \tag{40}$$

The first summation term can be bounded as $\frac{1}{4} \sum_{t=1}^{T} \|\mathbf{z}_t - \mathbf{z}_{t-1}\|^2 \leq \frac{1}{8(1-3\alpha)} D^2$ by Lemma C.2. For the second summation, note that $\lambda_t \leq \lambda_{t+1}$, we use Part (b) in Lemma C.4 to get

$$\sum_{t=1}^{T} \frac{1}{\lambda_t} \|\mathbf{F}(\mathbf{z}_t)\| \leq \frac{1}{\lambda_1} \|\mathbf{F}(\mathbf{z}_1)\| + \sum_{t=1}^{T-1} \left( \frac{1 + \alpha}{\eta_t} \|\mathbf{z}_{t+1} - \mathbf{z}_t\| + \frac{\alpha}{\eta_t} \|\mathbf{z}_t - \mathbf{z}_{t-1}\| \right) \tag{41}$$

Similarly, by using Cauchy-Schwarz inequalities, these lead to

$$\sum_{t=1}^{T} \frac{1}{\lambda_t} \|\mathbf{F}(\mathbf{z}_t)\| \leq \frac{1}{\lambda_1} \|\mathbf{F}(\mathbf{z}_1)\| + \frac{(1 + 2\alpha)D}{\sqrt{2(1 - 3\alpha)}} \sqrt{\sum_{t=1}^{T-1} \frac{1}{\eta_t^2}}. \tag{42}$$

Plugging this bound back in (40), we arrive at

$$\sum_{t=1}^{T} \frac{1}{\eta_t^2} \leq \frac{1}{8(1 - 3\alpha)} D^2 + \frac{1}{\alpha \lambda_1} \|\mathbf{F}(\mathbf{z}_1)\| + \frac{(1 + 2\alpha)D}{\alpha\sqrt{2(1 - 3\alpha)}} \sqrt{\sum_{t=1}^{T-1} \frac{1}{\eta_t^2}} \tag{43}$$

Note that $\sum_{t=0}^{T} \frac{1}{\eta_t^2}$ appears on both side of (37). Again, we apply Lemma B.5 to obtain the desired result

$$\sum_{t=1}^{T} \frac{1}{\eta_t^2} \leq \frac{1}{4(1 - 3\alpha)} D^2 + \frac{2}{\alpha \lambda_1} \|\mathbf{F}(\mathbf{z}_1)\| + \frac{(1 + 2\alpha)^2 D^2}{\alpha^2 (1 - 3\alpha)}$$

$\qquad\qquad\qquad\qquad\qquad\qquad\qquad\qquad\qquad\qquad\qquad\qquad\qquad\qquad\qquad\qquad\qquad\square$

We are finally at a position to prove the convergence theorem for the parameter-free algorithm, which is essentially a straightforward combination of the previous lemmas and propositions. Similar to the proof of the constant $\lambda$ setting, we accompany the convergence in the primal-dual gap with the complexity bound with respect to the norm of the operator (gradient of $f$). Due to space constraints, we present this complementary bound in the proof of the theorem.

**Proof of Theorem 6.2.** By Proposition C.3,

$$\mathrm{Gap}_{\mathcal{X}\times\mathcal{Y}}(\bar{\mathbf{z}}_{T+1}) \leq \max\{\lambda_1, L_2\}\left(\sup_{\mathbf{z}\in\mathcal{X}\times\mathcal{Y}}\|\mathbf{z}-\mathbf{z}^*\|^2 + \left(\frac{9}{8} + \frac{\alpha^2}{4(1-3\alpha)}\right)D^2\right)\left(\sum_{t=0}^{T}\eta_t\right)^{-1}.$$

Combining the above with the upper bound in Lemma C.5 completes the result.

Moreover, from (42), we also obtain that

$$\sum_{t=2}^{T+1}\frac{1}{\lambda_t}\|\mathbf{F}(\mathbf{z}_t)\| \leq \frac{(1+2\alpha)D}{\sqrt{2(1-3\alpha)}}\sqrt{\sum_{t=1}^{T}\frac{1}{\eta_t^2}}. \tag{44}$$

Combining (44) with Lemma C.5, we obtain that

$$\sum_{t=2}^{T+1}\frac{1}{\lambda_t}\|\mathbf{F}(\mathbf{z}_t)\| \leq \frac{(1+2\alpha)D}{\sqrt{2(1-3\alpha)}}\sqrt{\frac{17\alpha^2 + 16\alpha + 4}{4(1-3\alpha)\alpha^2}\|\mathbf{z}_1-\mathbf{z}^*\|^2 + \frac{2}{\alpha\lambda_1}\|\mathbf{F}(\mathbf{z}_1)\|}.$$

Finally, the result follows from the fact that $\min_{t\in\{2,...,T+1\}}\|\mathbf{F}(\mathbf{z}_t)\| \leq \frac{1}{T}\sum_{t=2}^{T+1}\|\mathbf{F}(\mathbf{z}_t)\|$.

$\square$

## C.1 Proof of Lemma C.1

We begin by formalizing the error condition implied by the parameter-free algorithm where $\lambda_t$ is chosen as in Option (II) in Step 4 in Algorithm 1.

**Lemma C.6.** *Consider the update rule in (6) and let $\lambda_t$ and $\eta_t$ be given by (8), respectively. Then we have $\eta_t\|\mathbf{z}_{t+1}-\mathbf{z}_t\| \leq 2\alpha$ and $\eta_t\|\mathbf{e}_{t+1}\| \leq \alpha\lambda_{t+1}\|\mathbf{z}_{t+1}-\mathbf{z}_t\|$.*

*Proof.* Similar to the proof of the analogous result in the constant $\lambda_t$ setting, note that $\eta_t$ is given as in (8) by solving the following quadratic equation:

$$\eta_t(\eta_t\|\mathbf{F}(\mathbf{z}_t)\| + \eta_{t-1}\|\mathbf{e}_t\|) = 2\alpha\lambda_t.$$

Thus, by using Lemma 5.2, we can prove that

$$\eta_t\|\mathbf{z}_{t+1}-\mathbf{z}_t\| \leq \frac{\eta_t}{\lambda_t}(\eta_t\|\mathbf{F}(\mathbf{z}_t)\| + \eta_{t-1}\|\mathbf{e}_t\|) \leq 2\alpha.$$

To prove the second inequality, note that by our choice of $\lambda_{t+1}$ in (8), it holds that $\lambda_{t+1} \geq \frac{2\|\mathbf{e}_{t+1}\|}{\|\mathbf{z}_{t+1}-\mathbf{z}_t\|^2}$ and thus $\|\mathbf{e}_{t+1}\| \leq \frac{\lambda_{t+1}}{2}\|\mathbf{z}_{t+1}-\mathbf{z}_t\|^2$. Hence, we also obtain $\eta_t\|\mathbf{e}_{t+1}\| \leq \frac{\lambda_{t+1}\eta_t}{2}\|\mathbf{z}_{t+1}-\mathbf{z}_t\|^2 \leq \alpha\lambda_{t+1}\|\mathbf{z}_{t+1}-\mathbf{z}_t\|$.

$\square$

Moving forward, we present the following upper bound on the approximation error using Assumption 2.3.

**Lemma C.7.** *Suppose that Assumption 2.3 holds. Then for any $t \geq 1$, we have*

$$\|\mathbf{e}_{t+1}\| := \|\mathbf{F}(\mathbf{z}_{t+1}) - \mathbf{F}(\mathbf{z}_t) - \mathbf{F}'(\mathbf{z}_t)(\mathbf{z}_{t+1}-\mathbf{z}_t)\| \leq 2L_1\|\mathbf{z}_{t+1}-\mathbf{z}_t\|.$$

*Proof.* By using the triangle inequality, we have $\|\mathbf{e}_{t+1}\| \leq \|\mathbf{F}(\mathbf{z}_{t+1})-\mathbf{F}(\mathbf{z}_t)\| + \|\mathbf{F}'(\mathbf{z}_t)(\mathbf{z}_{t+1}-\mathbf{z}_t)\|$. By Assumption 2.3, it holds that $\|\mathbf{F}(\mathbf{z}_{t+1})-\mathbf{F}(\mathbf{z}_t)\| \leq L_1\|\mathbf{z}_{t+1}-\mathbf{z}_t\|$ and $\|\mathbf{F}'(\mathbf{z}_t)\|_{\mathrm{op}} \leq L_1$. Hence, this further implies that $\|\mathbf{e}_{t+1}\| \leq \|\mathbf{F}(\mathbf{z}_{t+1})-\mathbf{F}(\mathbf{z}_t)\| + \|\mathbf{F}'(\mathbf{z}_t)\|\|\mathbf{z}_{t+1}-\mathbf{z}_t\| \leq 2L_1\|\mathbf{z}_{t+1}-\mathbf{z}_t\|$.

$\square$

**Proof of Lemma C.1.** Our starting point is the inequality (13) in Proposition 5.1. To begin with, we write

$$\frac{\eta_t}{\lambda_t}\langle\mathbf{e}_{t+1},\mathbf{z}_{t+1}-\mathbf{z}^*\rangle - \frac{\eta_{t-1}}{\lambda_t}\langle\mathbf{e}_t,\mathbf{z}_t-\mathbf{z}^*\rangle$$
$$= \frac{\eta_t}{\lambda_t}\langle\mathbf{e}_{t+1},\mathbf{z}_{t+1}-\mathbf{z}^*\rangle - \frac{\eta_{t-1}}{\lambda_{t-1}}\langle\mathbf{e}_t,\mathbf{z}_t-\mathbf{z}^*\rangle + \left(\frac{1}{\lambda_{t-1}}-\frac{1}{\lambda_t}\right)\eta_{t-1}\langle\mathbf{e}_t,\mathbf{z}_t-\mathbf{z}^*\rangle. \tag{45}$$

Note that the first two terms on the right-hand side of (45) telescope. Moreover, note that $\lambda_{t-1} \leq \lambda_t$ and thus $\frac{1}{\lambda_{t-1}} - \frac{1}{\lambda_t} \geq 0$. By using Lemma C.6, for $t \geq 2$ we can further bound

$$
\begin{aligned}
\left( \frac{1}{\lambda_{t-1}} - \frac{1}{\lambda_t} \right) \eta_{t-1} \langle \mathbf{e}_t, \mathbf{z}_t - \mathbf{z}^* \rangle & \leq \left( \frac{1}{\lambda_{t-1}} - \frac{1}{\lambda_t} \right) \eta_{t-1} \|\mathbf{e}_t\| \|\mathbf{z}_t - \mathbf{z}^*\| \\
& \leq \left( \frac{\lambda_t}{\lambda_{t-1}} - 1 \right) \alpha \|\mathbf{z}_t - \mathbf{z}_{t-1}\| \|\mathbf{z}_t - \mathbf{z}^*\| \\
& \leq \alpha^2 \|\mathbf{z}_t - \mathbf{z}_{t-1}\|^2 + \frac{1}{4} \left( \frac{\lambda_t}{\lambda_{t-1}} - 1 \right)^2 \|\mathbf{z}_t - \mathbf{z}^*\|^2.
\end{aligned}
\tag{46}
$$

Hence, by plugging in (46) in (45) and summing the inequality from $t = 1$ to $t = T$, we obtain that

$$
\begin{aligned}
\sum_{t=1}^{T} & \left( \frac{\eta_t}{\lambda_t} \langle \mathbf{e}_{t+1}, \mathbf{z}_{t+1} - \mathbf{z}^* \rangle - \frac{\eta_{t-1}}{\lambda_t} \langle \mathbf{e}_t, \mathbf{z}_t - \mathbf{z}^* \rangle \right) \\
& \leq \frac{\eta_T}{\lambda_T} \langle \mathbf{e}_{T+1}, \mathbf{z}_{T+1} - \mathbf{z}^* \rangle + \alpha^2 \sum_{t=2}^{T} \|\mathbf{z}_t - \mathbf{z}_{t-1}\|^2 + \frac{1}{4} \sum_{t=2}^{T} \left( \frac{\lambda_t}{\lambda_{t-1}} - 1 \right)^2 \|\mathbf{z}_t - \mathbf{z}^*\|^2,
\end{aligned}
$$

where we used the fact that $\eta_0 = 0$. Moreover, by Cauchy-Schwarz inequality, Lemma C.7, and Lemma C.6, we can bound

$$
\begin{aligned}
\frac{\eta_T}{\lambda_T} \langle \mathbf{e}_{T+1}, \mathbf{z}_{T+1} - \mathbf{z}^* \rangle & \leq \frac{\eta_T}{\lambda_T} \|\mathbf{e}_{T+1}\| \|\mathbf{z}_{T+1} - \mathbf{z}^*\| \leq \frac{2 L_1 \eta_T}{\lambda_T} \|\mathbf{z}_{T+1} - \mathbf{z}_T\| \|\mathbf{z}_{T+1} - \mathbf{z}^*\| \\
& \leq \frac{4 \alpha L_1}{\lambda_T} \|\mathbf{z}_{T+1} - \mathbf{z}^*\|.
\end{aligned}
$$

Furthermore, for the last error term in (13), we use Cauchy-Schwarz inequality, Lemma C.6, and Young's inequality to upper bound

$$
\begin{aligned}
\frac{\eta_{t-1}}{\lambda_t} \langle \mathbf{e}_t, \mathbf{z}_t - \mathbf{z}_{t+1} \rangle & \leq \frac{\eta_{t-1}}{\lambda_t} \|\mathbf{e}_t\| \|\mathbf{z}_t - \mathbf{z}_{t+1}\| \leq \alpha \|\mathbf{z}_t - \mathbf{z}_{t-1}\| \|\mathbf{z}_{t+1} - \mathbf{z}_t\| \\
& \leq \frac{\alpha}{2} \|\mathbf{z}_t - \mathbf{z}_{t-1}\|^2 + \frac{\alpha}{2} \|\mathbf{z}_{t+1} - \mathbf{z}_t\|^2.
\end{aligned}
$$

Combining all the inequalities above with (13) in Proposition 5.1, we arrive at

$$
\begin{aligned}
\frac{\|\mathbf{z}_{T+1} - \mathbf{z}^*\|^2}{2} \leq {} & \frac{\|\mathbf{z}_1 - \mathbf{z}^*\|^2}{2} - \sum_{t=1}^{T} \frac{\|\mathbf{z}_t - \mathbf{z}_{t+1}\|^2}{2} + \frac{4 \alpha L_1}{\lambda_T} \|\mathbf{z}_{T+1} - \mathbf{z}^*\| + \alpha^2 \sum_{t=2}^{T} \|\mathbf{z}_t - \mathbf{z}_{t-1}\|^2 \\
& + \frac{1}{4} \sum_{t=2}^{T} \left( \frac{\lambda_t}{\lambda_{t-1}} - 1 \right)^2 \|\mathbf{z}_t - \mathbf{z}^*\|^2 + \alpha \sum_{t=1}^{T} \|\mathbf{z}_{t+1} - \mathbf{z}_t\|^2.
\end{aligned}
$$

Since $\alpha < \frac{1}{2}$, we can bound $\alpha^2 < \frac{\alpha}{2}$. Rearranging the terms, we obtain

$$
\begin{aligned}
\frac{\|\mathbf{z}_{T+1} - \mathbf{z}^*\|^2}{2} - \frac{4 \alpha L_1}{\lambda_T} \|\mathbf{z}_{T+1} - \mathbf{z}^*\| \leq {} & \frac{\|\mathbf{z}_1 - \mathbf{z}^*\|^2}{2} + \frac{1}{4} \sum_{t=2}^{T} \left( \frac{\lambda_t}{\lambda_{t-1}} - 1 \right)^2 \|\mathbf{z}_t - \mathbf{z}^*\|^2 \\
& - \left( \frac{1 - 3\alpha}{2} \right) \sum_{t=1}^{T} \|\mathbf{z}_{t+1} - \mathbf{z}_t\|^2.
\end{aligned}
$$

Now we can complete the square and write the left-hand side as

$$
\begin{aligned}
\frac{1}{2} \|\mathbf{z}_{T+1} - \mathbf{z}^*\|^2 - \frac{4 \alpha L_1}{\lambda_T} \|\mathbf{z}_{T+1} - \mathbf{z}^*\| & = \frac{1}{2} \left( \|\mathbf{z}_{T+1} - \mathbf{z}^*\| - \frac{4 \alpha L_1}{\lambda_T} \right)^2 - \frac{8 \alpha^2 L_1^2}{\lambda_T^2} \\
& \geq \frac{1}{4} \|\mathbf{z}_{t+1} - \mathbf{z}^*\|^2 - \frac{16 \alpha^2 L_1^2}{\lambda_T^2},
\end{aligned}
$$

where we used the elementary inequality that $(a - b)^2 \geq \frac{1}{2} a^2 - b^2$. Combining the above two inequalities and changing $T$ to $t$ leads to the desired result.

## C.2  Proof of Lemma C.2

Define the auxiliary positive sequence $\{d_t\}_{t \geq 2}$ as follows:

$$d_2^2 = \frac{64\alpha^2 L_1^2}{\lambda_1^2} + 2\|\mathbf{z}_1 - \mathbf{z}^*\|^2, \quad d_{t+1}^2 = \frac{64\alpha^2 L_1^2}{\lambda_t^2} + 2\|\mathbf{z}_1 - \mathbf{z}^*\|^2 + \sum_{s=2}^{t} \left(\frac{\lambda_s}{\lambda_{s-1}} - 1\right)^2 d_s^2.$$

Then by using induction and Lemma C.1, we can easily prove that $\|\mathbf{z}_t - \mathbf{z}^*\| \leq d_t$ for all $t \geq 2$. Moreover, from the above recursive relation, for $t \geq 1$, we have

$$d_{t+1}^2 - d_t^2 = 64\alpha^2 L_1^2 \left(\frac{1}{\lambda_t^2} - \frac{1}{\lambda_{t-1}^2}\right) + \left(\frac{\lambda_t}{\lambda_{t-1}} - 1\right)^2 d_t^2.$$

Moreover, since $\lambda_t \geq \lambda_{t-1}$ by (8), we have

$$1 + \left(\frac{\lambda_t}{\lambda_{t-1}} - 1\right)^2 = \frac{\lambda_t^2}{\lambda_{t-1}^2} - 2\frac{\lambda_t}{\lambda_{t-1}} + 2 \leq \frac{\lambda_t^2}{\lambda_{t-1}^2}.$$

Hence, this implies that

$$d_{t+1}^2 \leq 64\alpha^2 L_1^2 \left(\frac{1}{\lambda_t^2} - \frac{1}{\lambda_{t-1}^2}\right) + \frac{\lambda_t^2}{\lambda_{t-1}^2} d_t^2$$

$$\Rightarrow \quad \frac{d_{t+1}^2}{\lambda_t^2} \leq \frac{d_t^2}{\lambda_{t-1}^2} + \frac{64\alpha^2 L_1^2}{\lambda_t^2} \left(\frac{1}{\lambda_t^2} - \frac{1}{\lambda_{t-1}^2}\right) \leq \frac{d_t^2}{\lambda_{t-1}^2} + \frac{64\alpha^2 L_1^2}{\lambda_1^2} \left(\frac{1}{\lambda_t^2} - \frac{1}{\lambda_{t-1}^2}\right).$$

By summing the above inequality from $t = 2$ to $t = T$, we obtain that

$$\frac{d_{T+1}^2}{\lambda_T^2} \leq \frac{d_2^2}{\lambda_1^2} + \frac{64\alpha^2 L_1^2}{\lambda_1^2} \left(\frac{1}{\lambda_T^2} - \frac{1}{\lambda_1^2}\right) = \frac{2\|\mathbf{z}_1 - \mathbf{z}^*\|^2}{\lambda_1^2} + \frac{64\alpha^2 L_1^2}{\lambda_1^2 \lambda_T^2}.$$

This implies that $d_{T+1}^2 \leq \frac{2\lambda_T^2}{\lambda_1^2}\|\mathbf{z}_1 - \mathbf{z}^*\|^2 + \frac{64\alpha^2 L_1^2}{\lambda_1^2}$. Since $\lambda_T \leq \max\{\lambda_1, L_1\}$, we obtain the final result.

Moreover, by rearranging the terms in (38), we also have

$$2(1 - 3\alpha) \sum_{s=0}^{t} \|\mathbf{z}_s - \mathbf{z}_{s+1}\|^2 \leq \frac{64\alpha^2 L_1^2}{\lambda_t^2} + 2\|\mathbf{z}_1 - \mathbf{z}^*\|^2 + \sum_{s=2}^{t} \left(\frac{\lambda_s}{\lambda_{s-1}} - 1\right)^2 \|\mathbf{z}_s - \mathbf{z}^*\|^2$$

$$\leq \frac{64\alpha^2 L_1^2}{\lambda_t^2} + 2\|\mathbf{z}_1 - \mathbf{z}^*\|^2 + \sum_{s=2}^{t} \left(\frac{\lambda_s}{\lambda_{s-1}} - 1\right)^2 d_s^2$$

$$= d_{t+1}^2 \leq D^2.$$

Dividing both sides by $2(1 - 3\alpha)$ finishes the proof.

## C.3  Proof of Proposition C.3

Our starting point is the inequality (9) in Proposition 5.1. To bound the first summation on the right-hand side, we write

$$\sum_{t=1}^{T} \frac{\lambda_t}{2} \left(\|\mathbf{z}_t - \mathbf{z}\|^2 - \|\mathbf{z}_{t+1} - \mathbf{z}\|^2\right) = \frac{\lambda_1}{2}\|\mathbf{z}_1 - \mathbf{z}\|^2 - \frac{\lambda_T}{2}\|\mathbf{z}_{T+1} - \mathbf{z}\|^2 + \sum_{t=2}^{T} \frac{\lambda_t - \lambda_{t-1}}{2}\|\mathbf{z}_t - \mathbf{z}\|^2$$

Moreover, since $\lambda_t \geq \lambda_{t-1}$ for any $t \geq 2$, we have

$$\sum_{t=2}^{T} \frac{\lambda_t - \lambda_{t-1}}{2}\|\mathbf{z}_t - \mathbf{z}\|^2 \leq \sum_{t=1}^{T} (\lambda_t - \lambda_{t-1}) \left(\|\mathbf{z}_t - \mathbf{z}^*\|^2 + \|\mathbf{z} - \mathbf{z}^*\|^2\right)$$

$$\leq \sum_{t=1}^{T} (\lambda_t - \lambda_{t-1}) \left(D^2 + \|\mathbf{z} - \mathbf{z}^*\|^2\right)$$

$$= (\lambda_T - \lambda_1) \left(D^2 + \|\mathbf{z} - \mathbf{z}^*\|^2\right).$$

Since $\frac{\lambda_1}{2}\|\mathbf{z}_1 - \mathbf{z}\|^2 \leq \lambda_1 \left(\|\mathbf{z}_1 - \mathbf{z}^*\|^2 + \|\mathbf{z} - \mathbf{z}^*\|^2\right) \leq \lambda_1 \left(D^2 + \|\mathbf{z} - \mathbf{z}^*\|^2\right)$, we obtain that $\sum_{t=1}^{T} \frac{\lambda_t}{2} \left(\|\mathbf{z}_t - \mathbf{z}\|^2 - \|\mathbf{z}_{t+1} - \mathbf{z}\|^2\right) \leq \lambda_T \left(D^2 + \|\mathbf{z} - \mathbf{z}^*\|^2\right) - \frac{\lambda_T}{2}\|\mathbf{z}_{T+1} - \mathbf{z}\|^2$.

Furthermore, we can use Lemma C.6 to control the error terms. By using the Cauchy-Swharz inequality and Young's inequality, for $t \geq 2$, we have

$$\eta_{t-1}\langle \mathbf{e}_t, \mathbf{z}_t - \mathbf{z}_{t+1}\rangle \leq \eta_{t-1}\|\mathbf{e}_t\|\|\mathbf{z}_t - \mathbf{z}_{t+1}\| \leq \frac{\eta_{t-1}^2}{2\lambda_t}\|\mathbf{e}_t\|^2 + \frac{\lambda_t}{2}\|\mathbf{z}_t - \mathbf{z}_{t+1}\|^2$$

$$\leq \frac{\alpha^2 \lambda_t}{2}\|\mathbf{z}_t - \mathbf{z}_{t-1}\|^2 + \frac{\lambda_t}{2}\|\mathbf{z}_t - \mathbf{z}_{t+1}\|^2$$

$$\leq \frac{\alpha^2 \lambda_T}{2}\|\mathbf{z}_t - \mathbf{z}_{t-1}\|^2 + \frac{\lambda_t}{2}\|\mathbf{z}_t - \mathbf{z}_{t+1}\|^2,$$

where we used Lemma C.6 in the second in the third inequality and the fact that $\{\lambda_t\}_{t\geq 0}$ is non-decreasing in the last inequality. By summing the above iequality from $t = 1$ to $t = T$, we obtain that

$$\sum_{t=1}^{T} \eta_{t-1}\langle \mathbf{e}_t, \mathbf{z}_t - \mathbf{z}_{t+1}\rangle \leq \frac{\alpha^2 \lambda_T}{2} \sum_{t=2}^{T} \|\mathbf{z}_t - \mathbf{z}_{t-1}\|^2 + \sum_{t=2}^{T} \frac{\lambda_t}{2}\|\mathbf{z}_t - \mathbf{z}_{t+1}\|^2$$

$$\leq \frac{\alpha^2 \lambda_T}{4(1 - 3\alpha)} D^2 + \sum_{t=2}^{T} \frac{\lambda_t}{2}\|\mathbf{z}_t - \mathbf{z}_{t+1}\|^2,$$

where we used Lemma C.2 in the last inequality. Similarly, using Cauchy-Schwarz and Young's inequalities, we can also bound

$$\eta_T\langle \mathbf{e}_{T+1}, \mathbf{z}_{T+1} - \mathbf{z}\rangle \leq \frac{\lambda_T}{2}\|\mathbf{z}_{T+1} - \mathbf{z}\|^2 + \frac{\eta_T^2}{2\lambda_T}\|\mathbf{e}_{T+1}\|^2.$$

Using Lemma C.7 and Lemma C.6, we further have $\frac{\eta_T^2}{2\lambda_T}\|\mathbf{e}_{T+1}\|^2 \leq \frac{2L_1^2}{\lambda_T}\eta_T^2\|\mathbf{z}_{T+1} - \mathbf{z}_T\|^2 \leq \frac{8\alpha^2 L_1^2}{\lambda_T}$. Combining all the inequalies above in (9), we obtain that

$$\sum_{t=1}^{T} \eta_t\langle \mathbf{F}(\mathbf{z}_{t+1}), \mathbf{z}_{t+1} - \mathbf{z}\rangle \leq \lambda_T \left(D^2 + \|\mathbf{z} - \mathbf{z}^*\|^2\right) - \frac{\lambda_T}{2}\|\mathbf{z}_{T+1} - \mathbf{z}\|^2 - \sum_{t=1}^{T} \frac{\lambda_t}{2}\|\mathbf{z}_t - \mathbf{z}_{t+1}\|^2$$

$$+ \frac{\lambda_T}{2}\|\mathbf{z}_{T+1} - \mathbf{z}\|^2 + \frac{8\alpha^2 L_1^2}{\lambda_T} + \frac{\lambda_T}{4(1 - 3\alpha)} D^2 + \sum_{t=2}^{T} \frac{\lambda_t}{2}\|\mathbf{z}_t - \mathbf{z}_{t+1}\|^2$$

$$\leq \lambda_T \|\mathbf{z} - \mathbf{z}^*\|^2 + \left(1 + \frac{\alpha^2}{4(1 - 3\alpha)}\right) \lambda_T D^2 + \frac{8\alpha^2 L_1^2}{\lambda_T}.$$

Finally, we used the fact that $\lambda_T \leq \max\{\lambda_1, L_2\}$ and $\frac{8\alpha^2 L_1^2}{\lambda_T} \leq \frac{8\alpha^2 L_1^2}{\lambda_1} \leq \frac{1}{8}\lambda_1 D^2 \leq \frac{1}{8}\lambda_T D^2$. The rest follows simiarly as in the proof of Proposition B.1.

## D    Omitted numerical experiments

**Implementation details:** We solve the linear systems in the subproblems of second-order methods exactly via MATLAB linear equation solver. The hyper-parameters for methods in the prior work are tuned to achieve the best performance per method. Specifically, for the HIPNEX method in [30], it has a hyper-parameter $\sigma \in (0, 0.5)$, which we choose in the interval $[0.05, 0.1, 0.15, \ldots, 0.45]$ for the best performance. Other hyper-parameters are determined by the formulas from [30]. For the Optimal SOM, the initial step size is set to be 1 as prescribed. Their algorithm has two line-search hyperparameters $\alpha$ and $\beta$. Note that their $\alpha$ is the same as ours, and we search for the best choice of $\alpha$ and $\beta$ for their algorithm from the interval $[0.1, 0.2, \ldots, 0.9]$. We use the combination that achieves the best empirical result.

Our first proposed algorithm in Option (**I**) does not require any tuning. We initialize our fully parameter-free method (Option (**II**)) using a heuristic, which essentially eliminates the tuning of its parameters. After we choose the initial point $\mathbf{z}_0$, we generate another random point $\hat{\mathbf{z}}_0$ which is

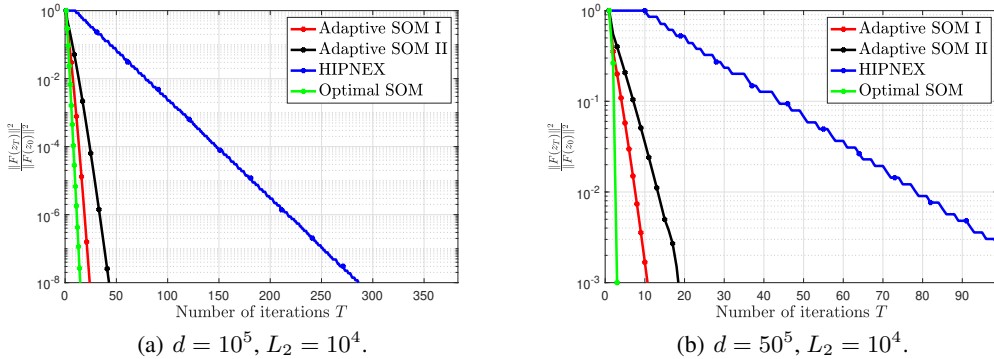

(a) $d = 10^5$, $L_2 = 10^4$.

(b) $d = 50^5$, $L_2 = 10^4$.

Figure 3: Synthetic min-max problem: convergence comparison with respect to iteration complexity.

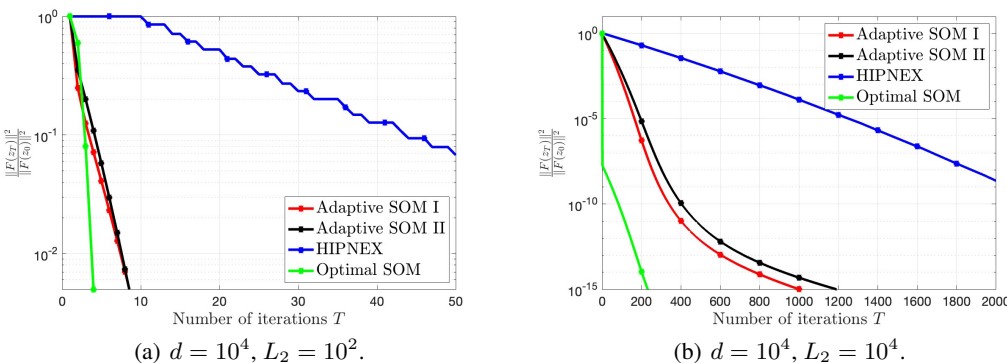

(a) $d = 10^4$, $L_2 = 10^2$.

(b) $d = 10^4$, $L_2 = 10^4$.

Figure 4: AUC maximization: convergence comparison with respect to iteration complexity.

close to the initial point. Then, using those two points we compute a local estimate to the Lipschitz constant $L_2$ for the values of $\lambda_0$ as $\lambda_0 = \frac{2\|\mathbf{F}(\hat{\mathbf{z}}_0) - \mathbf{F}(\mathbf{z}_0) - \mathbf{F}'(\mathbf{z}_0)(\hat{\mathbf{z}}_0 - \mathbf{z}_0)\|}{\|\hat{\mathbf{z}}_0 - \mathbf{z}_0\|^2}$. Empirically, we observe that this heuristic strategy is competitive and works well across different problem settings and instances.

Finally, we initialize all the algorithms at the same point $\mathbf{z}_0 = (\mathbf{x}_0, \mathbf{y}_0) \in \mathbb{R}^d$, drawn from the multivariate normal distribution.

**Additional experiments:** In Figures 3 and 4, we compare the performance among all 4 methods with respect to the iteration complexity for the synthetic min-max problem and the AUC maximization problem, respectively. Note that those plots do not account for the cost of linear search and backtracking; they are presented solely to complement Figures 1 and 2 for a complete comparison of the methods. For both problems, adaptive SOM I shows slightly better performance than Adaptive SOM II, consistent with the fact that Adaptive SOM I uses the exact Hessian Lipschitz parameter, while Adaptive SOM II estimates it. As expected, the Optimal SOM method, which uses a line search to pick the largest possible step size, has the best convergence. However, the performance of our adaptive line search-free method (Adaptive SOM I) and parameter-free method (Adaptive SOM II) is only slightly worse. Additionally, both of our methods outperform the HIPNEX method.

In Figure 5, we measure the runtime of the algorithms. When the dimension is small, the relative performance of the methods in terms of runtime is similar to that of in Figure 3 in terms of the number of iterations. On the other hand, in the high dimensional regime, the performance of the Optimal SOM becomes worse against other algorithms in the initial stage because they need to solve the linear equation multiple times during the backtracking line search scheme, which is computationally expensive and time-consuming when the dimension $d$ is large. Also observe that as the dimension increases, our methods perform gradually better than the line search-based approaches, which supports our claims on efficiency.

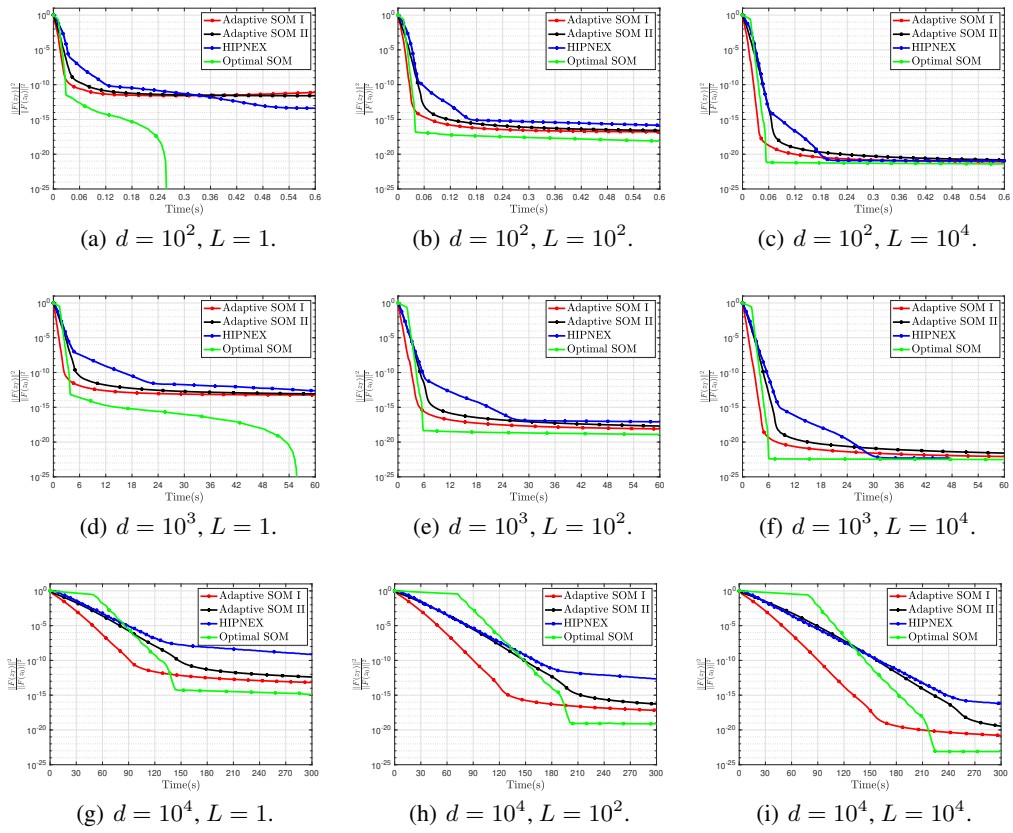

Figure 5: Synthetic min-max problem: additional plots for the convergence comparison with respect to runtime.

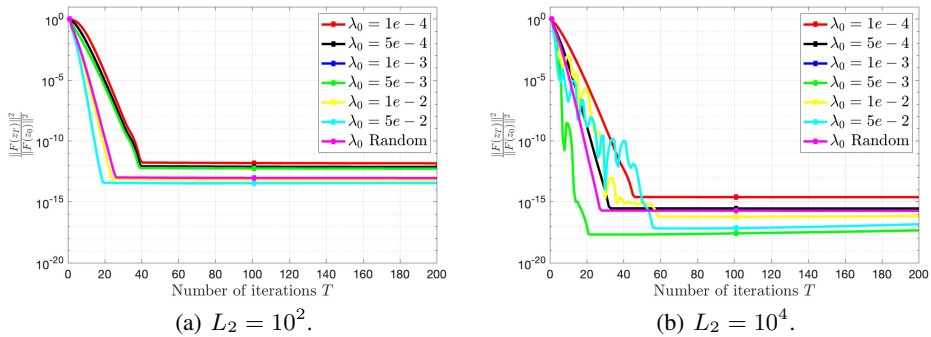

Figure 6: Runtime comparison for the parameter-free method (Option (**II**)) for solving the min-max problem in Section 7 ($d = 10^2$) with different initialization of $\lambda_0$.

As a complementary result, we tested the sensitivity of our parameter-free method (Option (**II**)) to the initialization of $\lambda_0$ and reported the results in Figure 6. Specifically, we considered the first min-max problem in Section 7, where $L_2 = 10^4$ and $d = 10^2$. Varying the initial choice of $\lambda_0$ from $10^{-4}$ to 0.05, Figure 6 shows that our method exhibits consistent performance. We also tested a heuristic initialization procedure as discussed above. The numerical results verify that our method is robust to initialization and our heuristic strategy is competitive and works well across different settings.

