# OpenReview forum: "Adaptive and Optimal Second-order Optimistic Methods for Minimax Optimization"
_NeurIPS.cc/2024/Conference — NeurIPS 2024 poster_

### Official Review · Reviewer_GGjT · 2024-07-09

**Soundness:** 4
**Presentation:** 3
**Contribution:** 4
**Rating:** 7
**Confidence:** 2

**Summary:**

This paper introduces adaptive, line search-free second-order methods aimed at solving convex-concave min-max problems. The proposed algorithms use an adaptive step size, simplifying the update rule to require solving only one linear system per iteration. The paper presents two main contributions: an adaptive second-order optimistic method that achieves an optimal convergence rate of O(1/T^1.5) and a parameter-free version that does not require knowledge of the Lipschitz constant of the Hessian. The algorithms are evaluated against existing methods, demonstrating practical efficiency and optimal rates.

**Strengths:**

1. The introduction of adaptive, line search-free second-order methods with optimal convergence rates for convex-concave min-max problems is a significant contribution to the field.
2. The paper provides both a parameter-free version and a version requiring minimal problem-specific information. The development of a parameter-free version of the algorithm that adapts based on local information without requiring the Lipschitz constant is noteworthy.
3. The contributions are clearly stated and proved to be the optimal through theoretical analysis and empirical results.

**Weaknesses:**

1. While the parameter-free method is innovative, it may still require careful tuning of initial parameters in practice.
2. Although the numerical experiment shows a promising result, it is limited to specific problem settings and may not generalize across diverse optimization tasks.

**Questions:**

1. How robust are the proposed methods to deviations from the assumed Lipschitz continuity of gradients and Hessians?
2. Can the parameter-free method's performance be significantly impacted by the choice of initial parameters?
3. How does the proposed approach compare to recent advances in machine learning applications, such as training GANs or solving reinforcement learning problems?

**Limitations:**

The authors pointed out the limitations that missing exact knowledge of the Lipschitz constant can slow down the convergence, and achieving the same parameter-free results without the assumption of Lipschitz continuous gradient remains an open problem.

---

> ### Author Rebuttal · Authors · 2024-08-07
>
> **W1 While the parameter-free method is innovative, it may still require careful tuning of initial parameters in practice.**
>
> **R1** This is an important observation, and thanks for bringing this point up. We agree that removing the necessary parameters will not come for free and we need to initialize $\lambda_1$ with an estimate of the Hessian's Lipschitz constant. Let us briefly discuss how the parameter-free method behaves for different initializations and also propose a simple initialization technique.
>
> Observe that $\eta_1 = O(\sqrt{\lambda_1})$ and $\eta_t = O(1 / \eta_{t-1})$ from Eq. (7). From this simplified view, we observe that the step sizes have a self-balancing property. Specifically, a large (or small) initial value of $\lambda_1$ will yield a large (or small) initial step size $\eta_1$. Yet, $\eta_2$ will be smaller (or larger) as it is inversely proportional to the **previous** step size. Eventually, the step size balances itself in both cases of initialization.
>
> To support our rationale, we would like to provide empirical verification. In Figure 3 in the rebuttal PDF, we run our parameter-free method for the same objective in Section 7 where $L_2 = 10^4$ and $d = 10^2$. The plot shows that our method has similar performance for a range of initialization.
>
> We also want to share the practical initialization we use in our experiments: we choose an initial point, $z_1$, and generate a second random point $\hat{z}_1$, close to $z_1$. Then, we compute the local $L_2$ estimate to initalize as $\lambda_1=2\\|F(\hat{z}_1)-F(z_1)-\nabla F(z_1) (\hat{z}_1 - z_1) \\|/\\|\hat{z}_1 - z_1\\|^2$. Let us clarify that this initialization rule comes from the proposed update recursion for $\lambda_t$ (in Alg. 1, line 4) and we use it in the experiments for consistency.
>
> ---
> **W2 Although the numerical experiment shows a promising result, ... may not generalize across diverse optimization tasks.**
>
> **R2** Thanks for raising this point. We ran a new set of experiments for the problem of maximizing an area under the receiver operating characteristic curve. This could be formulated as a min-max problem, where we want to find a classifier (set of weights) with **small** error that will also have a **large** area under the curve, as formulated in Eq 5.2 of [Lin et al., 2024].
>
> Please check out the respective plots under Figure 2 in the rebuttal PDF. Similar to the min-max problem in Section 7 of our paper, our methods converge relatively fast in the early stages of the execution. Due to time constraints, we have not run higher dimensional experiments for the new problem.
>
> ---
> **Q1 How robust are the proposed methods to deviations from the assumed Lipschitz continuity of gradients and Hessians?**
>
> **A1** This is an interesting question. Let us answer by focusing on the parameter-free algorithm (Option II).
>
> The Lipschitz gradient assumption is required **only** in the analysis to show that the iterates remain bounded, therefore, the algorithm is not affected by its variation. In fact, the experiments in our paper are based on an objective function which is not gradient Lipschitz but only Hessian Lipschitz, showing that our method is robust in that sense.
>
> We think our reviewer would agree that the main issue is the variation in the Hessian Lipschitz constant $L_2$. In Figure 3 of the Appendix, we varied the Hessian Lipschitz constant from $L = 1$ to $L = 10^4$. We noticed that our proposed method consistently performs well across these different settings and is competitive with the optimal SOM. Similar results can be seen in Figures 2 and 3 in our rebuttal PDF.
>
> ---
> **Q2 Can the parameter-free method's performance be significantly impacted by the choice of initial parameters?**
>
> **A2** Good question. We refer the reviewer to our response in **R1** above. As an additional note, a very small initial step size (corresponding to overshooting the Lipschitz constant $L_2$) will “delay” the achievement of the desired $O(1/T^{1.5})$, but our parameter-free method will eventually recover. When the initial estimate for $L_2$ is too small, our method will still converge as we explained in the previous answer, but non-parameter-free methods, which must know the exact Lipschitz constant, will diverge.
>
> ---
> **Q3 How does the proposed approach compare to recent advances in machine learning applications, such as training GANs or solving reinforcement learning problems?**
>
> **A3**  To begin with, we would like to acknowledge that second-order methods, both for minimization and min-max problems, might have limited use once the dimension of the data and the model increase, but we believe there are valid scenarios where second-order methods could be beneficial.
>
> Second-order methods can converge in fewer iterations, but calculating the Hessian and its inverse incurs extra costs per iteration. Thus, there is a trade-off between convergence rate and per-iteration cost. In scenarios where gradient evaluation is expensive, second-order methods may be more favorable than first-order methods, as they require fewer iterations and gradient queries.
>
> In terms of implementation efficiency, there are techniques to cut this cost significantly, such as computing the (inverse) Hessian-vector products (HVP). [Tran and Cutkosky, 2022] propose a second-order momentum method that uses Hessian-vector products for stochastic optimization. Their algorithm outperforms SGD and Adam in image and NLP tasks and it is only 1.3 to 1.7 times slower than SGD. In the reinforcement learning literature, [Salehkaleybar et al., 2022] developed a second-order policy gradient method, which outperforms baselines by a significant margin in terms of system probes. [Dagréou et al., 2024] conducted an empirical study of different HVP algorithms on Jax and showed that the cost of computing HVP is less than twice the cost of gradient computation. Thus, we may scale our method to high-dimensional problems similarly when combined with such techniques.

---

> ### Comment · Reviewer_GGjT · 2024-08-12
>
> Thanks for your constructive response that addresses my concerns. I will keep my score.
>
> Sincerely,
> Reviewer GGjT

---

### Official Review · Reviewer_w2ph · 2024-07-11

**Soundness:** 3
**Presentation:** 3
**Contribution:** 2
**Rating:** 6
**Confidence:** 2

**Summary:**

This paper proposes adaptive, line search-free second-order methods with an optimal rate of convergence for solving convex-concave min-max problems. By defining the step size recursively as a function of the gradient norm and the prediction error, they eliminate the need for line search or backtracking mechanisms. Additionally, the approach does not require knowledge of the Lipschitz constant of the Hessian.

**Strengths:**

1. The algorithms presented in the paper are novel. They eliminate the need for line search and backtracking by providing a closed-form, explicit, and simple iterate recursion with a data-adaptive step size.
2. The authors offer a clear explanation of how they developed this update.

**Weaknesses:**

The limitation of using line search is not clearly demonstrated. For example, it is unclear if using line search would incur higher computational costs and take more time. This limitation should be illustrated with experimental results. Specifically, in the numerical experiments, the Optimal SOM method shows the best convergence rate. Therefore, it is important to show how these novel algorithms outperform the Optimal SOM method.

**Questions:**

1.See Weakness above.
2. In Equations (6) and (7), \alpha appears in the update. However, \alpha is not mentioned in Algorithm 1. Can you explain its role in the algorithm? Besides, how is \alpha set in the experiments?
3. In the experiments, it is mentioned that “all the hyper-parameters are tuned to achieve the best performance per method.” Can you provide details on the tuning process for these hyper-parameters?

---

> ### Author Rebuttal · Authors · 2024-08-07
>
> **Q1 The limitation of using line search is not clearly demonstrated. For example, it is unclear if using line search would incur higher computational costs and take more time. This limitation should be illustrated with experimental results.**
>
> **A1** We believe our reviewer would agree with us that we cannot expect our method to beat the optimal SOM in terms of convergence rate (# of iterations needed to reach an accuracy). This is because the optimal SOM leverages the line search scheme to pick the largest possible step size, while our method achieves the same rate (up to constants) by **removing** the line search. Instead, our goal is to design a method that is easy to implement with minimal effect on performance.
>
> Indeed, there are scenarios where our method has better performance. Since our methods do not require line search, we expect them to be easier to implement and exhibit faster runtime, particularly when $L_2$ is large and, more importantly, when the problem is high-dimensional. When $L_2$ is large the line-search scheme might require several backtracking steps, and each step would be costly when $d$ is large due to the computation of Hessian and its inverse.
>
> Specifically, Figure 3 in our submission demonstrates the effect of increasing $d$ and $L_2$. Taking $10^{-15}$ as an acceptable accuracy, our methods reach the target faster than other methods when the problem has large $L_2$ and dimension. Having that said, we fully acknowledge your criticism that we should highlight the advantages of our linesearch-free design with more elaborate experiments.
>
> To this end, we have conducted new experiments with larger Lipschitz constant and higher dimensionality. We kindly ask you to check out **Figure 1** in the **rebuttal PDF** and the explanations we provided in the global response. With increasing dimensions and Lipschitz constant, we observe that our methods show significant gains against optimal SOM and HIPNEX.
>
> ---
> **Q2 In Equations (6) and (7), $\alpha$ appears in the update. However, $\alpha$ is not mentioned in Algorithm 1. Can you explain its role in the algorithm? Besides, how is $\alpha$ set in the experiments?**
>
> **A2** Thank you for raising this point. The reason that $\alpha$ does not appear in Algorithm 1 is that we set $\alpha = 0.25$ to simplify the expression. The parameter $\alpha$ stems from the condition in (4), and it controls the approximation error in the second-order optimistic method. Note that, unlike the Lipschitz constant, it is a **free** hyperparameter in our algorithm.  Our method with Option I needs $\alpha \in (0, 1/2)$, and the parameter-free version (Option II) requires $\alpha \in (0, 1/4)$. We can simply select it as $\alpha = 1/4$ to unify, which is what we do in the theorems and experiments.
>
> ---
> **Q3 In the experiments, it is mentioned that “all the hyper-parameters are tuned to achieve the best performance per method.” Can you provide details on the tuning process for these hyper-parameters?**
>
> **A3** Thanks for the question. For our first adaptive and line search-free second-order optimistic method, we simply choose $\lambda_t = L_2$. For the parameter-free method, the only hyper-parameter is $\lambda_0$, and we use a heuristic initialization for our method. We choose an initial point, $z_0$, and generate a second random point $\hat{z}_0$ which is close to $z_0$. Then, we compute the local $L_2$ estimate to initialize $\lambda_0 = 2 \\| F(\hat z_0) - F(z_0) - \nabla F(z_0) (\hat{z}_0 - z_0)  \\| / \\|\hat{z}_0 - z_0 \\|^2$. Note that we do not tune $\lambda_0$ as a hyper-parameter but use a simple initialization rule, which is in parallel with the proposed update recursion for $\lambda_t$ in Algorithm 1, line 4.
>
> For the HIPNEX method in [30], it has a hyperparameter $\sigma \in (0, 0.5)$, which we choose in the interval $[0.05, 0.1, 0.15, …, 0.45]$ for the best performance. Other hyper-parameters are determined by the formulas from the paper [30].
>
> For the Optimal SOM, the initial step size is set to be unit as prescribed. Their algorithm has two line search hyperparameters $\alpha, \beta$. Note that their $\alpha$ is the same as ours, and we search for the best choice of $\alpha$ and $\beta$ for their algorithm from the interval [0.1, 0.2, …, 0.9]. We use the combination that achieves the best empirical result.
>
> Thank you for raising this point and we will add the above details to the revision.

---

> > ### Comment · Reviewer_w2ph · 2024-08-08
> >
> > Thanks for your response. It addresses all my questions. And I'm willing to raise my score.
> >
> > Best.

---

### Official Review · Reviewer_gLyW · 2024-07-12

**Soundness:** 3
**Presentation:** 2
**Contribution:** 3
**Rating:** 5
**Confidence:** 3

**Summary:**

This paper consider solving convex-concave minmax problems using second order optimization method. A modified adaptive Optimistic algorithm by approximating the proximal point method using second order information is proposed. It is shown that this method can achieve the optimal convergence rate. Moreover, comparing with the previous second order methods, the method in this paper avoid the line search process and can avoid the precise knowledge of the Lipschitz constant used in finding suitable step sizes.

**Strengths:**

(1). The paper is well written. The basic mechanisms of the proposed algorithms and the difficulties of their analysis are clearly stated at the beginning of the paper.
(2). The motivation of this work that avoid the seemingly complex process of line search or knowledges of the Lipschitz constant is convinced. Moreover, the proposed algorithm, especially the (Option 2) method achieve this goal, although the additional assumption on the Lipschitz conditions on the gradient are assumed.

**Weaknesses:**

(1). Beside the theoretical advantages of the proposed methods and optimal SOM method, it is questionable that whether the proposed method has particle advantages compared to optimal SOM. From the experimental results, it seems both the convergence rate of optimal SOM (in Figure 1) and computational times in long term (in Figure 3) are better than the proposed adaptive SOM methods.

**Questions:**

(1). As in the weakness part, can the author provides more explains on the advantages of Adaptive SOM than optimal SOM?

(2). From Figure 2, it seems Adaptive SOM methods will not keep decreasing after some time node. Is this the case? What will happen if we add the iterations of optimization?

(3). From the work of (Mokhtari er al.,) both the optimistic and extra gradient methods can be put in a single framework of approximating the proximal point method. Can the authors provide some comments on the possibilities of using the second order approximate extra gradient method to construct algorithms?

**Limitations:**

There is no potential negative societal impact from this work.

---

> ### Author Rebuttal · Authors · 2024-08-07
>
> **Q1 Can the authors provide more explanation on the advantages of Adaptive SOM than optimal SOM?**
>
> **A1** In terms of convergence rate (# of iterations needed to reach an accuracy), one cannot expect our method to beat the optimal SOM. This is because the optimal SOM leverages the line search scheme to pick the largest possible step size, while our method achieves the same rate (up to constants) by **removing** the line search. Instead, our goal is to design a method that is easy to implement with minimal effect on performance.
>
> Indeed, there are scenarios where our method has better performance. Since our methods do not require line search, we expect them to be easier to implement and exhibit faster runtime, particularly when $L_2$ is large and, more importantly, when the problem is high-dimensional. When $L_2$ is large the line-search scheme might require several backtracking steps, and each step would be costly when $d$ is large due to the computation of Hessian and its inverse.
>
> Specifically, Figure 3 in our submission demonstrates the effect of increasing $d$ and $L_2$. Taking $10^{-15}$ as an acceptable accuracy, our methods reach the target faster than other methods when the problem has large $L_2$ and dimension. Furthermore, we have conducted new experiments with larger Lipschitz constant and higher dimensionality. Please check out Figure 1 in the rebuttal PDF and the explanations we provided in the global response. With increasing dimensions and Lipschitz constant, we observe that our methods show significant gains against optimal SOM and HIPNEX.
>
> ---
> **Q2 From Figure 2, it seems Adaptive SOM methods will not keep decreasing after some time node. Is this the case? What will happen if we add the iterations of optimization?**
>
> **A2** Let us highlight that Figure 2 studies the case where the Lipschitz constant is very small (L=1), which is in favor of the line-search methods. Optimal SOM can easily pick a large step size via the line search and converge faster, whereas our adaptive methods take a more conservative approach with smaller step sizes. Our methods will eventually reach the same error as optimal SOM with more iterations.
>
> Moreover, Figure 2 reports the performance against the number of iterations, which does not display the cost of line search. The negative effect of increasing the Lipschitz constant **in runtime** for optimal SOM can be observed through Figure 3 (g) -> (h) -> (i).
>
>
> ---
> **Q3 From the work of (Mokhtari et al.,2020) both the optimistic and extra gradient methods can be put in a single framework of approximating the proximal point method. Can the authors provide some comments on the possibilities of using the second-order approximate extra gradient method to construct algorithms?**
>
> **A3** This is an excellent question. Integrating our technique with the extragradient (EG) framework to remove the line search turns out to have some technical issues. Specifically, one of the fundamental components that help us avoid the line search is the data-adaptive recursion for $\eta_t$, and finding the respective formula for EG is the first challenge. Let us explain briefly without the proof details.
>
> EG computes an intermediate sequence, which we can call $z_{t+1/2}$, and uses the gradient information at this middle point to achieve the next point $z_{t+1}$. It is possible to propose a choice of $\eta_t$ that depends inversely on $\\|F(z_t)\\|$, which implies that, to lower bound $\eta_t$, we need to find an upper bound on $\\|F(z_t)\\|$. However, we are only able to upper bound $\\|F(z_{t+1/2})\\|$ instead of $\\|F(z_t)\\|$. This discrepancy prevents us from establishing the same convergence guarantee as in this paper. Nevertheless, this is a truly interesting question (also for minimization problems) and we are actively exploring whether a unification is possible.

---

### Official Review · Reviewer_4DVD · 2024-07-13

**Soundness:** 4
**Presentation:** 3
**Contribution:** 3
**Rating:** 7
**Confidence:** 4

**Summary:**

The authors present a new second-order method for convex-concave min-max optimization based on the optimistic gradient method but modified to work with second-order information. The authors first propose a variant that requires the value of Jacobian Lipschitzness $L_2$ and then introduce an additional parameter-free variant that does not require any hyperparameters. The proposed method obtains the optimal rate for this setting and appears to be quite practical. Finally, the authors test their method on a toy problem and compare it to other second-order methods for the same problem, where we can see the proposed method to perform the best.

**Strengths:**

The authors obtained the optimal rate for the studied problem with a practical method, which doesn't require solving complicated sub-problems. Furthermore, the authors designed a parameter-free version which appears to be very practical.

**Weaknesses:**

1. Unlike previous work on second-order methods for minimization and variational inequality, this work requires the gradients to be Lipschitz. I'm not even sure if that makes the method optimal in the class since usually the lower bounds are established without this extra assumption.
2. The problem might lack applications in machine learning. Min-max problems became popular recently due to a period of time when GANs were used, but it's a bit unrealistic to expect a second-order method to be useful in high-dimensional applications.
3. I think some related work is missing in terms of the algorithm design, in particular, error feedback papers (since this work uses error vectors) and papers on regularized Newton methods (since this work also tries to eliminate line search).
4. Some more practical experiments would have been welcome here.
## Minor
The light green in the plots is very difficult to see with the white background, I'd suggest changing the colors in the plots based on modern standards for figure formatting
In Section 7, vectors $x$ and $y$ should be made bold to be in line with the formatting style of the rest of the paper
Some equations, in the main body and in the appendix, are missing punctuation, e.g., lines 178, 203, 207, 452, 454, 488, etc.

**Questions:**

1. I do not see any mention of the error feedback technique, but the method resembles it a lot. Did it motivate the authors in any way?
2. Can a faster convergence rate of $O(1/T^2)$ be established for the same method when we specialize the problem to minimizing a convex function, as has been established for regularized Newton methods?
3. Why do you need to study specifically min-max problems?
4. Is Assumption 2.3 needed only to bound the iterate norms?
5. You assume the gradients to be Lipschitz, doesn't it mean that the $O(1/T^{1.5})$ lower bounds that you cite are no longer applicable since they are established for the class of operators with Lipschitz Jacobian?

**Limitations:**

I find the limitations section to be right on point.

---

> ### Author Rebuttal · Authors · 2024-08-07
>
> **W1 This work requires Lipschitz gradients thus lower bounds are not applicable.**
>
> **R1** First, we would like to highlight that our first method (Option I) does not require Lipschitz gradients and achieves the optimal rate for its setting. However, the parameter-free version (Option II) does require the Lipschitz gradients. Consequently, we agree that the existing lower bounds do not apply when Assumption 2.3 is in play. We will add a paragraph to clarify this.
>
> However, we would like to propose educated guesses based on analogous lower bounds in convex minimization. Arjevani et al. (2019) shows that for convex minimization with Lipschitz gradient and Hessian, the optimal rate is $O(\min\\{T^{-2},T^{-3.5}\\} )$. Since the lower bound for minimizing a convex function with only Lipschitz Hessian is $O(1/T^{3.5})$, the Lipschitz gradient assumption does not improve the lower bound. Similarly, we believe this additional assumption for min-max optimization would not lead to a rate improvement. We will include a remark in our revised version.
>
> ---
> **W2 Second-order methods may not be useful in high-dimensional applications.**
>
> **R2** We acknowledge that second-order methods might have limited use as the dimension of the data and the model increase. However, there are scenarios where they could be beneficial.
>
> Second-order methods can converge in fewer iterations, though they incur the extra cost of calculating the Hessian and its inverse. Thus, there is a trade-off between convergence rate and per-iteration cost. In cases where gradient evaluation is expensive, second-order methods may be more favorable than first-order methods, as they require fewer iterations and gradient queries.
>
> To improve implementation efficiency, techniques like computing the (inverse) Hessian-vector products (HVP) can significantly reduce costs. [Dagréou et al., 2024] showed that the cost of HVP is less than twice the cost of gradient computation. Thus, we may scale our method to high-dimensional problems when combined with such techniques - please also check out our response **A3** to **Reviewer GGjT**.
>
> ---
> **W3 Some related work is missing.**
>
> **R3** Regarding regularized Newton’s methods, we covered the relevant papers for min-max optimization to our knowledge [18,19,20,21,22,23]. We understand that our reviewer might also refer to regularized methods for convex minimization. We only cite the core work [29], and we will happily add the following on (accelerated) cubic methods [Nesterov, 2008; Jiang et al., 2020] and quadratic regularization variants [Mischenko, 2021; Antonakopoulos et al., 2022].
>
> Regarding error feedback, we were not initially aware of this line of work and built our method purely based on optimistic methods, which have been studied since 1980 [9] and have become popular recently [12, 14, 15]. However, we agree one could draw a high-level connection. Focusing on the pioneering work [Seide et al., 2014], error feedback algorithms keep an aggregated vector of errors from an “approximation” step such as compression/quantization. This **aggregated** error is added every iteration to correct the update of the local point. We highlight that our optimistic approach focuses only on the difference between the current and the last iterate, while error feedback vectors accumulate the entire history of errors. We will include a discussion on error feedback algorithms and their relevance. Thank you for making us aware of this.
>
> ---
> **W4 Practical experiments.**
>
> **R4** We have included a new experiment for the problem of maximizing the area under the ROC curve. We also repeated the experiment in our paper with higher dimensions and larger Lipschitz constants. Our new plots indicate a clear advantage for our algorithms compared to optimal SOM and HIPNEX once the dimension and $L_2$ increase. Please check out our new plots in the rebuttal PDF and the discussions in our global response.
>
> ---
> **Q1 Error feedback technique.**
>
> **A1** Please check our response in **R3**.
>
> ---
> **Q2 Can a faster rate of $O(1/T^2)$ be established for minimizing a convex function?**
>
> **A2** This is an excellent question. To our knowledge, no second-order optimistic method for convex minimization, even with line search, achieves a rate better than $O(1/T^{1.5})$. Our current analysis gives this same rate for the convex minimization setting. However, quadratic regularization of Newton’s method with adaptive regularization [Mishchenko, 2022] and cubic regularization of Newton’s method [29] achieve a faster rate of $O(1/T^2)$. Therefore, we conjecture that a minor modification of our method should achieve the same rate. This is a direction we are currently pursuing.
>
> ---
> **Q3 Why study Min-Max?**
>
> **A3** Min-max problems have been studied for several decades, long before the popularization of GANs. They have been explored in various formulations, such as variational inequalities, across fields like game theory, economics, and multi-agent learning. With the recent interest in second and higher-order methods, many open problems in min-max optimization need new algorithmic designs. We believe our adaptive parameter strategies offer a solid step forward by providing a new framework to bypass line search, applicable to minimization algorithms as well. We will include a thorough discussion on min-max problems, their relevance, and the implications of our design for other fields.
>
> ---
> **Q4 Assumption 2.3.**
>
> **A4** The reviewer is correct: this assumption is only needed to ensure iterates remain bounded. Note that many papers on parameter-agnostic algorithms **artificially** assume bounded iterates (see [32, 33,34, 35, Antonakopoulos et al., 2022]). We addressed this by replacing boundedness with the milder assumption of Lipschitz gradients. Although this complicates the analysis, we believe it is an important step forward.
>
> ---
> **Q5 The $O(1/T^{1.5})$ lower bounds are not applicable.**
>
> **A5** Please check our response in **R1**.

---

> > ### Comment · Reviewer_4DVD · 2024-08-07
> >
> > Thanks for your response.
> >
> > W1. Thank you for the clarification regarding the assumptions, I did miss that Assumption 2.3 is not needed for the first method.
> >
> > I'm not sure I understand your argument regarding the lower bounds. As you pointed out, the lower bound of Arjevani et al. includes a $1/T^2$ term. If the gradients are Lipschitz with a small constant and the Hessians are Lipschitz with a large one, the $1/T^2$ term can be much smaller. This is especially realistic since the Lipschitzness of the Hessians is a strictly stronger assumption on any bounded set. I'd suggest the authors refrain from any big claims on optimality of their methods when Lipschitz gradients are assumed.
> >
> > W2. I cannot agree with your argument about the Hessian-vector product. If we're using backpropagation, why should we compute it for the quadratic approximation of the problem (to solve the Newton iteration) instead of computing it for the problem itself? In my experience, Newton-like methods are only useful when we can use efficient linear algebra solvers for the arising linear systems, while in situations where we use backpropagation, they only introduce extra hyperparameters.
> >
> > W3. I'm surprised the authors were not familiar with the error-feedback literature as you even used the same notation $e_t$ for the error term, though I realize it is the natural choice. I think it's worth citing a paper on the topic since the connection is so strong, with the mention that you designed your method independently.
> >
> > W4. I hoped the authors would find an example from some recent NeurIPS papers concerned with applications where minmax problems needed to be solved, to provide an interesting example of minmax problem and test the methods on it. I realize the method is unlikely to be useful for GAN training, but if there are no relevant problems at all, it raises again the question of how relevant the designed methods are to NeurIPS.
> >
> > Q2. Thanks for the response, I'm looking forward to the new method.
> >
> > Q3. I apologize for formulating my initial question so loosely, I did not mean to question your interest in minmax problems, though it was interesting to read your response to this question. I meant to ask why your theory does not give us guarantees for the more general problem of monotone inclusion, which generalizes unconstrained minmax optimization.

---

> ### Author Response · Authors · 2024-08-08
>
> Thank you for reading our rebuttal and for sharing your additional comments.
>
> **W1 Arguments regarding the lower bounds.**
>
> Sorry for the confusion. Indeed, the lower bound presented in Arjevani et al. (2023) takes the form $\Omega(\min \\{ L_1D^2/T^2, L_2 D^3/T^{3.5}\\})$, where $L_1$ is the gradient Lipschitz constant, $L_2$ is the Hessian Lipschitz constant, and $D = \\|z_0 -z^*\\|$ is the initial distance to the optimum. As the reviewer correctly points out, determining which of the two bounds is the minimum depends on the Lipschitz constants and the initial distance. However, for sufficiently large $T$, the latter bound, $L_2 D^3/T^{3.5}$, will eventually become the smaller one, implying that the optimal dependence on $T$ is $1/T^{3.5}$.
> Likewise, our hypothesis is that the lower bound for min-max optimization with Lipschitz gradients and Hessians would follow a similar structure given by $\Omega(\min \\{ L_1D^2/T, L_2 D^3/T^{1.5}\\})$. If that is indeed the case, we can argue that our convergence rate is optimal in terms of the dependence on $T$. We will make sure to clarify this nuance in our revision and provide the necessary context. Thank you for your insightful question.
>
> ---
> **W2 Arguments about the Hessian-vector product.**
>
> We apologize if our arguments on Hessian-vector products (HVPs) caused any confusion. Our reviewer has a valid point that computing gradients with backpropagation is generally more cost-effective than dealing with HVPs. However, it is also possible to compute HVPs effectively using the backpropagation framework (there are Pytorch packages specifically developed for this purpose), thereby avoiding explicit matrix inversions. Having that said, we acknowledge that this may not be suitable in all scenarios. We intended to highlight examples, such as Tran and Cutkosky (2022), which demonstrate efficient implementations of second-order methods using HVPs.
>
> Finally, we would like to remark that our focus is mainly on the theoretical aspects of optimization methods. Nonetheless, we acknowledge that making our method more practical is an important direction for future research.
>
> Hoang Tran and Ashok Cutkosky. Better SGD using Second-order Momentum. NeurIPS 2022.
>
> ---
> **W3 The error-feedback literature.**
>
> Thank you for again making us aware of this line of work; we will include a discussion on error feedback in our revision. Also, please feel free to suggest any specific references that you have in mind beyond Seide et al. (2014).
>
> Frank Seide, et al. "1-bit stochastic gradient descent and its application to data-parallel distributed training of speech DNNs." Interspeech 2014.
>
> ---
> **W4 I hoped the authors provide an interesting example of min-max problems and test the methods on it.**
>
> Thank you for your suggestion. Another possible application of min-max optimization in machine learning is robust adversarial training (Tsipras et al., 2018; Madry et al., 2018), where the goal is to train a classifier that is robust to adversarial perturbations. This problem is indeed of high interest in machine learning and appears in several applications. Moreover, Javanmard et al. (2018) showed that in the special case of linear regression, the adversarial training problem is equivalent to a convex-concave min-max problem, which satisfies the assumptions in our paper. If the reviewer finds it necessary, we would be happy to test our proposed algorithms on this problem and report the numerical results here.
>
>
> Dimitris Tsipras, Shibani Santurkar, Logan Engstrom, Alexander Turner, and Aleksander Madry. Robustness may be at odds with accuracy. arXiv preprint 2018.
>
> Aleksander Madry, Aleksandar Makelov, Ludwig Schmidt, Dimitris Tsipras, and Adrian Vladu. Towards deep learning models resistant to adversarial attacks. ICLR 2018.
>
> Adel Javanmard, Mahdi Soltanolkotabi, and Hamed Hassani. Precise tradeoffs in adversarial training for linear regression. COLT 2020.
>
> ---
> **Q3 Why your theory does not give us guarantees for the more general problem of monotone inclusion, which generalizes unconstrained min-max optimization.**
>
> Thank you for the clarification. In fact, our convergence results can be extended to the more general problem of monotone inclusion $0 \in F(z) + H(z)$, with some proper modification to the algorithm. For instance, instead of using the operator norm $\\|F(z_k)\\|$ in our step size rule (7), we will use $\\|F(z_k) + v_k\\|$, where $v_k$ is a specific element in $H(z_k)$ that we construct from the algorithm. In our submission, we chose to focus on an unconstrained min-max problem for the ease of presentation, so that we can better highlight the key novelty of our techniques and make it accessible to a broader audience. We are planning to add this additional result in the appendix.

---

> > ### Comment · Reviewer_4DVD · 2024-08-14
> >
> > Thanks for the additional input.
> >
> > For the lower bounds, just please be precise when making statements about optimal rates.
> >
> > The paper that you mention is a good reference for error feedback.
> >
> > You don't have to study adversarial training numerically, I think your paper can be accepted as is with the theoretical focus.
> >
> > I agree the more general setting of monotone inclusion can be discussed in the appendix, though I think it's worth mentioning that your theory can be extended somewhere in the main body.

---

### Author Rebuttal · Authors · 2024-08-07

We thank the reviewers for their insightful feedback. Following your suggestions, we have performed new experiments, and the plots are included in the shared PDF file.

- **Practical advantage compared to optimal SOM**. To demonstrate the computational efficiency of our proposed line-search-free method, we consider the same min-max problem in Section 7 with a higher dimension and larger Lipschitz constant. We observe from Figure 1 that our line-search-free methods and HIPNEX in [30] consistently outperform the optimal SOM in terms of runtime. Moreover, the performance gap widens as the dimension of the problem increases; the line-search scheme requires several backtracking steps, especially when $L_2$ is larger, and each step would be costly due to the computation of Hessian and its inverse when $d$ is large. Additionally, both of our methods outperform the HIPNEX method.
- **Application to AUC maximization problems**. We consider a new problem of maximizing an area under the receiver operating characteristic curve. This could be formulated as a min-max problem, where we want to find a classifier (set of weights) with small error that will also have a large area under the curve, as formulated in Eq 5.2 of [Lin et al., 2024]. Similar to the observations above, Figure 2 demonstrates that both of our methods outperform the optimal SOM and HIPNEX in terms of runtime, particularly in the early stages of the execution.
- **The impact of the initial parameter $\lambda_0$**. We tested our parameter-free method on the same min-max problem in Section 7 where $L_2 = 10^4$ and $d = 10^2$. Varying the initial choice of $\lambda_0$ from $10^{-4}$ to $0.05$, Figure 3 shows that our method exhibits consistent performance. We also tested a heuristic initialization procedure used in our other experiments ("$\lambda_0$ random" in the figure). Specifically, we choose an initial point, $z_0$, and generate a second random point $\hat{z}_0$ close to $z_0$. Then, we compute the local $L_2$ estimate to initalize $\lambda_0=2\\|F(\hat{z}_0)-F(z_0)-\nabla F(z_0) (\hat{z}_0 - z_0) \\|/\\|\hat{z}_0 - z_0\\|^2$. We also observe that this heuristic strategy is competitive and works well across different settings.

We will include these new experiments and the above discussions in our revision.


---
**Additional references in our rebuttal:**

Hoang Tran, Ashok Cutkosky. Better SGD using Second-order Momentum. NeurIPS 2022.

Salehkaleybar, S., Khorasani, S., Kiyavash, N., He, N., & Thiran, P. Momentum-Based Policy Gradient with Second-Order Information, 2022.

Mathieu Dagréou, Pierre Ablin, Samuel Vaiter, Thomas Moreau. How to compute Hessian-vector products? ICLR Blogposts 2024.

Yossi Arjevani, Ohad Shamir, and Ron Shiff. Oracle complexity of second-order methods for smooth convex optimization. Mathematical Programming, 2019

Yuri Nesterov. Accelerating the cubic regularization of Newton’s method on convex problems. Mathematical Programming, 2008.

Bo Jiang, Tianyi Lin, and Shuzhong Zhang. A unified adaptive tensor approximation scheme to accelerate composite convex optimization. SIAM Journal on Optimization, 2020.

Konstantin Mishchenko. Regularized Newton method with global $O(1/k^2)$ convergence, 2021.

Kimon Antonakopoulos, Ali Kavis, and Volkan Cevher. Extra-newton: A first approach to noise-adaptive accelerated second-order methods. NeurIPS, 2022.

R. Monteiro and B. F. Svaiter. An accelerated hybrid proximal extragradient method for convex optimization and its implications to second-order methods. SIAM Journal on Optimization, 2013.

Tianyi Lin, Panayotis Mertikopoulos, Michael Jordan. Explicit Second-Order Min-Max Optimization Methods with Optimal Convergence Guarantee. 2024

---

### Decision · Program_Chairs · 2024-09-25

**Decision:**

Accept (poster)

**Comment:**

The paper presents a new second-order method for convex-concave min-max optimization and obtain the optimal rate for the studied problem. Moreover, the proposed algorithm appears to be very practical. All the reviewers agreed that the work contributed to the field. I suggest that the authors consider the changes proposed by the reviewers in their revision.